



# Converting Snow Depth to Snow Water Equivalent Using Climatological Variables

David F. Hill[1], Elizabeth A. Burakowski[2], Ryan L. Crumley[3], Julia Keon[4], J. Michelle Hu[5], Anthony A. Arendt[6], Katreen Wikstrom Jones[7], Gabriel J. Wolken[8]

[1]Civil and Construction Engineering, Oregon State University, OR, USA
[2]Institute for the Study of Earth, Oceans, and Space, University of New Hampshire, NH, USA
[3]Water Resources Graduate Program, Oregon State University, OR, USA
[4]Civil and Construction Engineering, Oregon State University, OR, USA
[5]Civil and Environmental Engineering, University of Washington
[6]Applied Physics Laboratory, University of Washington
[7]Alaska Division of Geological & Geophysical Surveys, Fairbanks, AK, USA
[8]Alaska Division of Geological & Geophysical Surveys, Fairbanks, AK, USA; International Arctic Research Center, University of Alaska Fairbanks, Fairbanks, AK, USA

*Correspondence to*: David F. Hill (david.hill@oregonstate.edu)

**Abstract.** We present a simple method that allows snow depth measurements to be converted to snow water equivalent (SWE) estimates. These estimates are useful to individuals interested in water resources, ecological function, and avalanche forecasting. They can also be assimilated into models to help improve predictions of total water volumes over large regions. The conversion of depth to SWE is particularly valuable since snow depth measurements are far more numerous than costlier and more complex SWE measurements. Our model regresses SWE against snow depth and climatological (30-year normal) values for mean annual precipitation (*MAP*) and mean February temperature, producing a power-law relationship. Relying on climatological normals rather than weather data for a given year allows our model to be applied at measurement sites lacking a weather station. Separate equations are obtained for the accumulation and the ablation phases of the snowpack, which introduces 'day of water year' (DOY) as an additional variable. The model is validated against a large database of snow pillow measurements and yields a bias in SWE of less than 0.5 mm and a root-mean-squared-error (RMSE) in SWE of approximately 65 mm. When the errors are investigated on a station-by-station basis, the average RMSE is about 5% of the *MAP* at each station. The model is additionally validated against a completely independent set of data from the northeast United States. Finally, the results are compared with other models for bulk density that have varying degrees of complexity and that were built in multiple geographic regions. The results show that the model described in this paper has the best performance for the validation data set.



## 1 Introduction

In many parts of the world, snow plays a leading-order role in the hydrological cycle (Mote et al., 2018). Accurate information about the spatial and temporal distribution of snow water equivalent (SWE) is useful to many stakeholders (water resource planners, avalanche forecasters, aquatic ecologists, etc.), but can be time consuming and expensive to obtain.

Snow pillows (Beaumont, 1965) are a well-established tool for measuring SWE at fixed locations. Figure 1 provides a conceptual sketch of the variation of SWE with time over a typical water year. A comparatively long accumulation phase is followed by a short ablation phase. While simple in operation, snow pillows are relatively large in size and they need to be installed prior to the onset of the season's snowfall. This limits their ability to be rapidly or opportunistically deployed. Additionally, snow pillow installations tend to require vehicular access, limiting their locations to relatively simple topography, and are not representative of the lowest or highest elevation bands within mountainous regions (Molotch and Bales, 2005). In the western United States (USA), the Natural Resources Conservation Service (NRCS) operates a large network of Snow Telemetry (SNOTEL) sites, featuring snow pillows. The NRCS also operates the smaller Soil Climate Analysis Network (SCAN) which provides the only, and very limited, snow pillow SWE measurements in the eastern USA.

SWE can also be measured manually, using a snow coring device that measures the weight of a known volume of snow to determine snow density (Church, 1933). These measurements are often one-off measurements, or in the case of 'snow courses' they are repeated weekly or monthly at a given location. The simplicity and portability of these devices expand the range over which measurements can be collected, but it can be challenging to apply these methods to deep snowpacks due to the length of standard coring devices and/or the need to dig very deep snowpits. Studies comparing different styles of snow samplers report statistically different results, suggesting that SWE measurements are sensitive to the design of the coring device, such as the presence of holes or slots, the device material, etc. (Beaumont and Work, 1963; Dixon and Boon, 2012).

Finally, SWE can be estimated with remote sensing methods, including satellite, airborne, and fixed platforms (e.g., Sokol et al., 2003; Vuyovich et al., 2014). Microwave frequencies are commonly used, but these frequencies do not work well in the presence of liquid water in the snowpack (Leinss et al., 2015). Recent attention has focused on the superior ability of L-band frequencies to measure SWE in wet snowpacks. Kang and Barros (2011) developed and tested an L-band snow sensor system in laboratory conditions and Deeb et al. (2017) discuss the application of L-band measurements to field-scale snow depth and SWE estimates for the SnowEx project.

All methods of measuring SWE are challenged by the fact that SWE is a depth-integrated property of a snowpack. This is why the snowpack must be weighed, in the case of a snow pillow, or a core must be extracted from the surface to the ground. This measurement complexity makes it difficult to obtain SWE information with the spatial and temporal resolution desired for watershed-scale studies. Other snowpack properties, such as the depth $h$, are

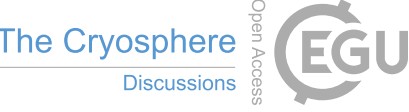



much easier to measure. Using a graduated device such as a meterstick or an avalanche probe to measure the depth
takes only seconds, and depth measurements at a fixed location are easily automated using low-cost ultrasonic
devices (Ryan et al., 2008). High-spatial-resolution measurements of snowpack depth are commonly made with
Light Detection and Ranging (LIDAR). One example of this is the Airborne Snow Observatory program (ASO;
Painter et al., 2016). The comparatively high expense of airborne LIDAR surveys typical limits measurements
geographically (to a few basins) and temporally (weekly to monthly interval).

Given the relative ease in obtaining depth measurements, it is common to use $h$ as a proxy for SWE. Figure 1 shows
a conceptual sketch of the variation of SWE with $h$ over a typical water year. Noting the arrows on the curve, we see
that SWE is multi-valued for each $h$. This is due to the fact that the snowpack increases in density throughout the
water year, producing a hysteresis loop in the curve. A large body of literature exists on the topic of how to convert
$h$ to SWE. It is beyond the scope of this paper to provide a full review of these 'bulk density equations,' where the
density is given by $\rho_b = \text{SWE}/h$. Instead, we refer readers to the useful comparative review by Avanzi et al. (2015).
Here, we prefer to discuss a limited number of previous studies that illustrate the spectrum of methodologies and
complexities that can be used to determine $\rho_b$ or SWE.

Many studies express $\rho_b$ as an increasing function (often linear) of $h$. In some cases (e.g., Lundberg et al., 2006) a
second equation is added where $\rho_b$ attains a constant value when a threshold $h$ is exceeded. A single linear equation
captures the process of densification of the snowpack during the accumulation phase, but performs poorly during the
ablation phase, where depths are decreasing but densities continue to increase or approach a constant value.
Other approaches choose to parameterize $\rho_b$ in terms of time, rather than $h$. Pistocchi (2016) provides a single
equation while Mizukami and Perica (2008) provide two sets of equations, one set each for early and late season.
Each set contains four equations, each of which is applicable to a particular 'cluster' of stations. This clustering was
driven by observed densification characteristics and the resulting clusters are relatively spatially discontinuous.
Jonas and Magnusson (2009) take the idea of region- (or cluster-) specific equations and extend it further to provide
coefficients that depend on time and elevation as well. They use a simple linear equation for $\rho_b$ in terms of $h$ and the
slope and intercept of the equation are given as monthly values, with three elevation bins for each month (36 pairs of
coefficients). There is an additional contribution to the intercept (or 'offset') which is region-specific (one of 7
regions).

These classifications, whether based on region, elevation, or season, are valuable since they acknowledge that all
snow is not equal. Sturm et al. (2010) address this directly by developing a snow density equation where the
coefficients depend upon the 'snow class' (5 classes). Sturm et al. (1995) explain the decision tree, based on
temperature, precipitation, and wind speed, that leads to the classification. The temperature metric is the 'cooling
degree month' calculated during winter months only. Similarly, only precipitation falling during winter months was
used in the classification. Finally, given the challenges in obtaining high quality, high-spatial-resolution wind



information, vegetation classification was used as a proxy. Using climatological values (rather than values for a
given year), Sturm et al. (1995) were able to develop a global map of snow classification.

There are many other formulations for snow density that increase in complexity and data requirements. Meloysund
et al. (2007) express $\rho_b$ in terms of sub-daily measurements of relative humidity, wind characteristics, air pressure,
and rainfall, as well as $h$ and estimates of solar exposure ('sun hours'). McCreight and Small (2014) use daily snow
depth measurements to develop their regression equation. They demonstrate improved performance over both Sturm
et al. (2010) and Jonas and Magnusson (2009). However, a key difference between the McCreight and Small (2014)
model and the others listed above is that the former cannot be applied to a single snow depth measurement. Instead,
it requires a continuous time series of depth measurements at a fixed location. Further increases in complexity (and
correspondingly, accuracy) are found in energy-balance snowpack models (SnowModel, Liston and Elder, 2006;
VIC, Liang et al., 1994, DHSVM, Wigmosta et al., 1994, others). While the particular details vary, these models
generally require high temporal-resolution time series of many meteorological variables as input. Also, many of
these models resolve vertical variations in snow density which are wholly absent from the bulk (vertically averaged)
density approaches reviewed above.

Despite the development of multi-layer energy-balance snow models, there is still a demonstrated need for bulk
density formulations and for vertically integrated data products like SWE. Pagano et al. (2009) review the
advantages and disadvantages of energy-balance models and statistical models and describe how the NRCS uses
SWE (from SNOTEL stations) and accumulated precipitation in their statistical models to make daily water supply
forecasts. If SWE information is desired at a location that does not have a SNOTEL station, and if not part of a
modeling effort, then bulk density equations and depth measurements are an excellent choice.

The present paper seeks to generalize the ideas of Mizukami and Perica (2008), Jonas and Magnusson (2009), and
Sturm et al., (2010). Specifically, our goal is to regress physical and environmental variables directly into the
equations. In this way, environmental variability is handled in a continuous fashion rather than in a discrete way
(model coefficients based on classes). The main motivation for this comes from evidence (e.g., Fig. 3 of Alford,
1967) that density can vary significantly over short distances on a given day. Bulk density equations that rely solely
on time completely miss this variability and equations that have coarse (model coefficients varying over either
vertical bins or horizontal grids) spatial resolution may not fully capture it either.

Our approach is most similar to Mizukami and Perica (2008), Jonas and Magnusson (2009), and Sturm et al., (2010)
in that a minimum of information is needed for the calculations; we intentionally avoid approaches like Meloysund
et al. (2007) and McCreight and Small (2014). This is because our interests are in converting $h$ measurements to
SWE estimates in areas lacking weather instrumentation. The following sections introduce the numerous data sets
that were used in this study, outline the regression model adopted, and assess the performance of the model.





## 2 Methods

### 2.1 Data

#### 2.1.1 Snow Depth and Snow Water Equivalent

In this section, we list sources of 1970-present snow data utilized for this study (Table 1).

##### 2.1.1.1 USA NRCS Snow Telemetry and Soil Climate Analysis Networks

SNOTEL (Serreze et al., 1999; Dressler et al., 2006) and SCAN (Schaefer et al. 2007) stations in the contiguous United States (CONUS) and Alaska typically record sub-daily observations of $h$, SWE, and a variety of weather variables (Figure 2a-b). The periods of record are variable, but the vast majority of stations have a period of record in excess of 30 years. For this study, data from all SNOTEL sites in CONUS and Alaska and northeast USA SCAN sites were obtained with the exception of sites whose period of record data were unavailable online. Only stations with both SWE and $h$ data were retained.

##### 2.1.1.2 Canada (British Columbia) Snow Survey Data

Goodison et al. (1987) note that Canada has no national digital archive of snow observations from the many independent agencies that collect snow data and that snow data are instead managed provincially. The quantity and availability of the data vary considerably among the provinces. The Water Management Branch of the British Columbia (BC) Ministry of the Environment manages a comparatively dense network of Automated Snow Weather Stations (ASWS) that measure SWE, $h$, accumulated precipitation, and other weather variables (Figure 2a). For this study, data from all British Columbia ASWS sites were initially obtained. As with the NRCS stations, only ASWS stations with both SWE and $h$ data were retained.

##### 2.1.1.3 Northeast USA Data

Snow data for this project from the northeast US come from two networks and three research sites (Figure 2b). The Maine Cooperative Snow Survey (MCSS, 2018) network includes $h$ and SWE data collected by the Maine Geological Survey, the United States Geological Survey, and numerous private contributors and contractors. MCSS snow data are collected using the Standard Federal or Adirondack snow sampling tubes typically on a weekly to bi-weekly schedule throughout the winter and spring, 1951-present. The New York Snow Survey network data were obtained from the National Oceanic and Atmospheric Administration's Northeast Regional Climate Center at Cornell University (NYSS, 2018). Similar to the MCSS, NYSS data are collected using Standard Federal or Adirondack snow sampling tubes on weekly to bi-weekly schedules, 1938-present.

The Sleepers River, Vermont Research Watershed in Danville, Vermont (Shanley and Chalmers, 1999) is a USGS site that includes 15 stations with long-term weekly records of $h$ and SWE collected using Adirondack snow tubes. Most of the periods of record are 1981-present, with a few stations going back to the 1960s. The sites include



topographically flat openings in conifer stands, old fields with shrub and grass, a hayfield, a pasture, and openings in
mixed softwood-hardwood forests. The Hubbard Brook Experiment Forest (Campbell et al., 2010) has collected
weekly snow observations at the Station 2 rain gauge site, 1959-present. Measurement protocol collects ten samples
2 m apart along a 20 m transect in a hardwood forest opening about ¼ hectare in size. At each sample location along
the transect, $h$ and SWE are measured using a Mt. Rose snow tube and the ten samples are averaged for each
transect. Finally, the Thompson Farm Research site includes a mixed hardwood forest site and an open pasture site
(Burakowski et al. 2013; Burakowski et al. 2015). Daily (from 2011-2018), at each site, a snow core is extracted
with an aluminum tube and weighed (tube + snow) using a digital hanging scale. The net weight of the snow is
combined with the depth and the tube diameter to determine $\rho_b$, similar to a Federal or Adirondack sampler.

**2.1.1.4 Chugach Mountains (Alaska) Data**
In the spring of 2018, we conducted three weeks of fieldwork in the Chugach mountains in coastal Alaska, near the
city of Valdez (Figure 2c-d). We measured SWE using a Federal sampler at 71 sites along elevational transects
during March, April, and May. The elevational transects ranged between 250 and 1100 m (net change along
transect) and were accessible by ski and snowshoe travel. At each of those 71 sites, we took 3 SWE and $h$
measurements within 1 m² and averaged the result. Additionally, we used an avalanche probe to measure $h$ in 8
locations within the surrounding 10 m², resulting in a total of 550+ snow depth measurements. These 71 sites were
scattered across 8 regions in order to capture spatial gradients in snow densities that exist in the Chugach mountains
as the wetter, more-dense maritime snow near the coast gradually changes to drier, less dense snow on the interior
side.

**2.1.1.5 Outlier Detection and Removal**
Figure 3 demonstrates that it is not uncommon for automated snow depth measurements to become noisy or non-
physical, at times reporting large depths when there is no SWE reported. It was therefore desirable to apply some
objective, uniform procedure to each station's dataset in order to remove clear outlier points. We recognize that
there is no accepted standardized method for cleaning bivariate SWE-$h$ data sets. While Serreze et al. (1999) offer a
procedure for SNOTEL data in their appendix, it is relevant only for precipitation and SWE values, not $h$. Given the
strong correlation between $h$ and SWE, we instead choose to use common outlier detection techniques for bivariate
data.

The Mahalanobis distance (MD; Maesschalck et al., 2000) quantifies how far a point lies from the mean of a
bivariate distribution. The distances are in terms of the number of standard deviations along the respective principal
component axes of the distribution. For highly correlated bivariate data, the MD can be qualitatively thought of as a
measure of how far a given point deviates from an ellipse enclosing the bulk of the data. One problem is that the MD
is based on the statistical properties of the bivariate data (mean, covariance) and these properties can be adversely
affected by outlier values. Therefore, it has been suggested (e.g., Leys et al., 2018) that a 'robust' MD (RMD) be
calculated. The RMD is essentially the MD calculated based on statistical properties of the distribution unaffected





by the outliers. This can be done using the Minimum Covariance Determinant (MCD) method as first introduced by
Rousseeuw (1984).

Once RMDs have been calculated for a bivariate data set, there is the question of how large an RMD must be in
order for the data point to be considered an outlier. For bivariate normal data, the distribution of the square of the
RMD is $\chi^2$ (Gnanadesikan and Kettenring, 1972), with p (the dimension of the dataset) degrees of freedom. So, a
rule for identifying outliers could be implemented by selecting as a threshold some arbitrary quantile (say 0.99) of
$\chi_p^2$. For the current study, a threshold quantile of 0.999 was determined to be an appropriate compromise in terms of
removing obviously outlier points, yet retaining physically plausible results.

A scatter plot of SWE vs. $h$ for the uncleaned SNOTEL dataset from CONUS and AK reveals many non-physical
points, mostly when a very large $h$ is reported for a very low SWE (Figure 4a). Approximately 0.7% of the original
data points were removed in the cleaning process described above, creating a more physically plausible scatter plot
(Figure 4b). Note that the outlier detection process was applied to each station individually. The same procedure was
applied to the BC and northeast USA data sets as well (not shown). Table 1 summarizes useful information about
the numerous data sets described above and indicates the final number of data points retained for each.

Table 1: Summary of information about the datasets used in this study. The numbers of stations and data points
reflect the post-processed data.

| Dataset Name | Dataset Type | Number of retained stations | Number of retained data points | Precision ($h$ / SWE) |
|---|---|---|---|---|
| NRCS SNOTEL NRCS SCAN | Snow pillow (SWE), ultrasonic ($h$) | 791 5 | 1,900,000 7094 | (0.5 in / 0.1 in) (0.5 in / 0.1 in) |
| British Columbia Snow Survey | Snow pillow (SWE), ultrasonic ($h$) | 31 | 61,000 | (1 cm / 1 mm) |
| Maine Geological Survey | Adirondack or Federal sampler (SWE and $h$) | 431 | 28,000 | (0.5 in / 0.5 in ) |
| Hubbard Brook (Station 2), NH | Mount Rose sampler (SWE and $h$) | 1 | 704 | (0.1 in / 0.1 in) |
| Thompson Farm, NH | Snow core (SWE and $h$) | 2 | 988 | 0.5 in / 0.5 in) |
| Sleepers River, VT | Adirondack sampler | 14 | 7214 | (0.5 in / 0.5 in) |
| New York Snow Survey | Adirondack or Federal sampler (SWE and $h$) | 523 | 44,614 | (0.5 in / 0.5 in) |
| Chugach Mountains, AK | Federal sampler (SWE and $h$) and avalanche probe ($h$) | 71 | 71 | (0.5 in / 0.5 in) for sampler; 1 cm for probe |


**2.1.2 Climatological Variables**
30-year climate normals at 800 m (nominal) resolution for CONUS and for the period 1981-2010 were obtained
from the PRISM website (Daly et al., 1994). PRISM normals for British Columbia (BC), Canada, were obtained
from the ClimateBC project (Wang et al., 2012), also for the 1981-2010 period. Finally, PRISM normals for Alaska
(AK) were obtained from the Integrated Resource Management Applications (IRMA) Portal run by the National



Park Service. The AK normals are for the 1971-2000 period and have a slightly coarser resolution (approximately
1.5 km). Figure 5 shows gridded maps of mean annual precipitation ($MAP$) and mean February Temperature ($\bar{T}_F$)
for these three climate products, plotted together. Other temperature products (max and min temperatures; other
months) were obtained as well, but are not shown.

**2.2 Regression Model**
In order to demonstrate the varying degrees of influence of explanatory variables, several regression models were
constructed. In each case, the model was built by randomly selecting 50% of the paired SWE-$h$ measurements from
the aggregated CONUS, AK, and BC snow pillow datasets. The model was then validated by applying it to the
remaining 50% of the dataset and comparing the modeled SWE to the observed SWE for those points. Additional
validation was done with the northeast USA datasets which were completely left out of the model building process.

**2.2.1 One-Equation Model**
The simplest equation, and one that is supported by the strong correlation seen in Figure 3, is one that expresses
SWE as a function of $h$. A linear model is attractive in terms of simplicity, but this limits the snowpack to a constant
density. An alternative is to express SWE as a power law, i.e.,

(1)      $SWE = Ah^{a_1}$.

This equation can be log-transformed into

(2)      $log_{10}(SWE) = log_{10}(A) + a_1 log_{10}(h)$

which immediately allows for simple linear regression methods to be applied. With both $h$ and SWE expressed in
units of mm, the obtained coefficients are $(A, a_1) = (0.146, 1.102)$. Information on the performance of the model
will be deferred until the results section.

**2.2.2 Two-Equation Model**
Recall from Figures 1 and 4 that there is a hysteresis loop in the SWE-$h$ relationship. During the accumulation
phase, snow densities are relatively low. During the ablation phase, the densities are relatively high. So, the same
snowpack depth is associated with two different SWEs, depending upon the time of year. The regression equation
given above does not resolve this difference. This can be addressed by developing two separate regression
equations, one for the accumulation ($acc$) and one for the ablation ($abl$) phase. This approach takes the form

(3)      $SWE_{acc} = Ah^{a_1}; \quad DOY < DOY^*$

(4)      $SWE_{abl} = Bh^{b_1}; \quad DOY \geq DOY^*$




where $DOY$ is the number of days from the start of the water-year (October 1 is the origin), and $DOY^*$ is the critical
or dividing day-of-water-year separating the two phases. Put another way, $DOY^*$ is the day of peak SWE.
Interannual variability results in a range of $DOY^*$ for a given site. Additionally, some sites, particularly the SCAN
sites in the northeast USA, demonstrate multi-peak SWE profiles in some years. To reduce model complexity,
however, we investigated the use of a simple climatological (long term average) value of $DOY^*$. For each snow
pillow station, the average $DOY^*$ was computed over the period of record of that station. Analysis of all of the
stations revealed that this average $DOY^*$ was relatively well correlated with the climatological mean April maximum
temperature (the average of the daily maximums recorded in April; $R^2 = 0.7$). However, subsequent regression
analysis demonstrated that the SWE estimates were relatively insensitive to $DOY^*$ and the best results were actually
obtained when $DOY^*$ was uniformly set to 180 for all stations. Again, with both SWE and $h$ in units of mm, the
regression coefficients turn out to be $(A, a_1) = (0.150, 1.082)$ and $(B, b_1) = (0.239, 1.069)$.

As these two equations are discontinuous at $DOY^*$, they are blended smoothly together to produce the final two-
equation model

(5)      $SWE = SWE_{acc}\frac{1}{2}(1 - tanh[0.01\{DOY - DOY^*\}]) +$
$$SWE_{abl}\frac{1}{2}(1 + tanh[0.01\{DOY - DOY^*\}])$$

The coefficient 0.01 in the tanh function controls the width of the blending window and was selected to minimize
the root mean square error of the model estimates.

**2.2.3 Two-Equation Model with Climate Parameters**
A final model was constructed by incorporating climatological variables. Again, the emphasis is this study is on
methods that can be implemented at locations lacking the time series of weather variables that might be available at
a weather or SNOTEL station. Climatological normals are unable to account for interannual variability, but they do
preserve the high spatial gradients in climate that can lead to spatial gradients in snowpack characteristics. Stepwise
linear regression was used to determine which variables to include in the regression. The initial list of potential
variables included was

(6)      $SWE = f\left(h, z, MAP, \bar{T}_{J_{min}}, \bar{T}_{J_{mean}}, \bar{T}_{J_{max}}, \bar{T}_{F_{min}}, \bar{T}_{F_{mean}}, \bar{T}_{F_{max}}, \bar{T}_{M_{min}}, \bar{T}_{M_{mean}}, \bar{T}_{M_{max}}, \bar{T}_{A_{min}}, \bar{T}_{A_{mean}}, \bar{T}_{A_{max}}\right)$

where $z$ is the elevation (m), $MAP$ is the mean annual precipitation (mm) and the temperatures (°$C$) represent the
mean of minimum, mean, and maximum daily values for the months January through April (J, F, M, A). For
example, $\bar{T}_{J_{min}}$ is the climatological normal of the average of the daily minimum temperatures observed in January.





In the stepwise regression, explanatory variables were accepted if they improved the adjusted $R^2$ value by 0.001.
The result of the regression yielded

(7)     $SWE_{acc} = Ah^{a_1}MAP^{a_2}\left(\bar{T}_{F_{mean}} + 30\right)^{a_3}; \quad DOY < DOY^*$

(8)     $SWE_{abl} = Bh^{b_1}MAP^{b_2}\left(\bar{T}_{F_{mean}} + 30\right)^{b_3}; \quad DOY \geq DOY^*$

or, in log-transformed format,

(9)     $log_{10}(SWE_{acc}) = log_{10}(A) + a_1 log_{10}(h) +$

325                        $a_2 log_{10}(MAP) + a_3 log_{10}\left(\bar{T}_{F_{mean}} + 30\right); \quad DOY < DOY^*$


(10)    $log_{10}(SWE_{abl}) = log_{10}(B) + b_1 log_{10}(h) +$

328                        $b_2 log_{10}(MAP) + b_3 log_{10}\left(\bar{T}_{F_{mean}} + 30\right); \quad DOY \geq DOY^*$


indicating that only snow depth, mean annual precipitation and mean February temperature were relevant. Manual
tests of model construction with other variables included confirmed that Eqns. (7-8) yielded the best results. In the
above equations, note that an offset is added to the temperature in order to avoid taking the log of a negative
number. These two SWE estimates for the individual (*acc* and *abl*) phases of the snowpack are then blended with
Eqn. (5) to produce a single equation for SWE spanning the entire water year. The obtained regression coefficients
were $(A, a_1, a_2, a_3) = (0.0128, 1.070, 0.132, 0.506)$ and $(B, b_1, b_2, b_3) = (0.0271, 1.038, 0.201, 0.310)$. The
physical interpretation of these coefficients is straightforward. The fact that the coefficients on depth are greater than
unity indicates that the density (SWE/h) increases as the snowpack depth increases. The positive coefficients
associated with $MAP$ and $\bar{T}_{F_{mean}}$ indicate that snow densities should be higher in warmer, wetter locations than in
colder, drier locations.
**3 Results**
A comparison of the three regression models (one-equation model, Eq. (2); two-equation model, Eqs. (3-5); multi-
variable two-equation model, Eqs. (5, 7-8)) is provided in Figure 6. The left column shows scatter plots of modeled
SWE to observed SWE for the validation data set with the 1:1 line shown in black. The right column shows
histograms of the model residuals. The vertical lines in the right column show the mean error, or model bias.
Visually, it is clear that the one-equation model performs relatively poorly with a large negative bias. This is easily
explained. The SNOTEL measurements are uniformly spaced in time (daily). Given that the accumulation season is
much longer than the ablation season (Figure 1), there are many more data points that are representative of the
accumulation season. The model fit is weighted towards these points, which leads to large negative residuals in the
ablation season. This large negative bias is partially overcome by the two-equation model (middle row, Figure 6).
The cloud of points is closer to the 1:1 line and the vertical black line indicating the mean error is closer to zero. In





the final row of Figure 6, we see that the multi-variable two-equation model yields the best result by far. The
residuals are now evenly distributed with a negligible bias. Several metrics of performance for the three models,
including $R^2$ (Pearson coefficient), bias, and root-mean-square-error (RMSE), are provided in Table 2.

Table 2: Summary of performance metrics for the three regression models presented in Section 2.2.

| Model | $R^2$ | Bias (mm) | RMSE (mm) |
|---|---|---|---|
| One-equation | 0.946 | -19.5 | 102 |
| Two-equation | 0.962 | -5.1 | 81 |
| Multi-variable two-equation | 0.972 | -0.5 | 67 |


Model errors will have varying impact on the local snow regime depending on the total precipitation in a specific
region. Therefore, an RMSE was computed at each station location and normalized by the PRISM estimate of *MAP*
at that location. Figure 7 shows the probability density function of these normalized errors. The average RMSE is
approximately 5% of *MAP*, with most falling into the range of 2-8%. The spatial distribution of these normalized
errors is shown in Figure 8. For the SNOTEL stations, there are no clear regional patterns governing the normalized
errors, with the possible exception of the Sierra Nevada, where the errors are consistently higher than elsewhere.
The British Columbia stations also show higher overall errors.

**3.1 Results for Snow Classes**
A key objective of this study is to regress climatological information in a continuous rather than a discrete way. The
work by Sturm et al. (2010) therefore provides a valuable point of comparison. In that study, the authors developed
the following equation for density $\rho_b$

(11)      $\rho_b = (\rho_{max} - \rho_0)\left[1 - e^{(-k_1 h - k_2 DOY)}\right] + \rho_0$

where $\rho_0$ is the initial density, $\rho_{max}$ is the maximum or 'final' density (end of water year), $k_1$ and $k_2$ are coefficients,
and DOY in this case begins on January 1. This means that their DOY for October 1 is -92. The coefficients vary
with snow class and the values determined by Sturm et al. (2010) are shown in Table 3.

Table 3: Model parameters by snow class for Sturm et al. (2010).

| Snow Class | $\rho_{max}$ | $\rho_0$ | $k_1$ | $k_2$ |
|---|---|---|---|---|
| Alpine | 0.5975 | 0.2237 | 0.0012 | 0.0038 |
| Maritime | 0.5979 | 0.2578 | 0.0010 | 0.0038 |
| Prairie | 0.5941 | 0.2332 | 0.0016 | 0.0031 |
| Tundra | 0.3630 | 0.2425 | 0.0029 | 0.0049 |
| Taiga | 0.2170 | 0.2170 | 0.0000 | 0.0000 |






To make a comparison, the snow class for each SNOTEL (including CONUS, AK, and BC) site was determined
using a 1-km snow class grid (Sturm et al., 2010) and Equation (11) was used to estimate snow density (and then
SWE) for every point in the validation dataset described in Section 2.2. Figure 9 compares the SWE estimates from
the Sturm model and from the present multi-variable, two-equation model (Equations 5, 7-8). The upper left panel of
Figure 9 shows all of the data, and the remaining panels show the results for each snow class. In all cases, the
current model provides better estimates. Plots of the residuals by snow class are provided in Figure 10, giving an
indication of the bias of each model for each snow class. Summaries of the model performance, broken out by snow
class, are given in Table 4.

Table 4: Comparison of model performance by Sturm et al. (2010) and the present study.

| Model | Sturm et al. (2010) | | | Multi-variable two-equation model | | |
|---|---|---|---|---|---|---|
| Snow Class | $R^2$ | Bias (mm) | RMSE (mm) | $R^2$ | Bias (mm) | RMSE (mm) |
| All Data | 0.928 | -29.2 | 111 | 0.972 | -0.5 | 67 |
| Alpine | 0.973 | 10.1 | 55 | 0.971 | -0.3 | 55 |
| Maritime | 0.968 | -16.8 | 109 | 0.970 | -4.5 | 105 |
| Prairie | 0.967 | 18.7 | 56 | 0.965 | -0.2 | 51 |
| Tundra | 0.956 | -10.5 | 82 | 0.969 | -6.1 | 67 |
| Taiga | 0.943 | -80.0 | 151 | 0.971 | 2.4 | 62 |


**3.2 Results for Northeast USA**
The regression equations in this study were developed using a large collection of SNOTEL sites in CONUS, AK,
and BC. The snow pillow sites are limited to locations west of approximately W 105° (Figure 2a). By design, the
data sets from the northeastern USA (Section 2.1.1.3) were left as an entirely independent validation set. These
northeastern sites are geographically distant from the training data sets, are subject to a very different climate, and
are generally at much lower elevations than the western sites, providing an interesting opportunity to test how robust
the present model is.

Figure 11 graphically summarizes the datasets and the performance of the multi-variable two-equation model of the
current study. The RMSE values are comparable to those found for the western stations, but, given the
comparatively thinner snowpacks in the northeast, represent a larger relative error (Table 5). The bias of the model
is consistently positive, in contrast to the western stations where the bias was negligible.

Table 5: Performance metrics for the multi-variable two-equation model applied to various northeastern USA
datasets.

| Dataset Name | $R^2$ | Bias (mm) | RMSE (mm) |
|---|---|---|---|
| Maine Geological Survey, ME | 0.91 | 8.9 | 33.3 |
| Hubbard Brook (Station 2), NH | 0.63 | 18.9 | 64.2 |
| Thompson Farm, NH | 0.85 | 7.1 | 21.6 |
| NRCS SCAN | 0.87 | -1.8 | 38.7 |
| Sleepers River, VT | 0.93 | 14.0 | 29.7 |
| New York Snow Survey | 0.93 | 13.8 | 31.2 |




### 3.3 Results for Chugach Mountains

The results for the Federal sampler core measurements in the Chugach Mountains are shown in Figure 12, using a
format consistent with Figure 11. The three different measurement campaigns (March, April, and May) can be seen
by the different symbol colors in the left panel. One notable difference between Figures 11 and 12 is that the
Chugach dataset only spans spring months and not the full water year. So, the cluster of data points does not start at
the origin. The R2, bias (mm) and RMSE (mm) are 0.89, -50.0 and 118.0, respectively.

### 4 Discussion

The results presented in this study show that the regression equation described by equations (5, 7-8) is an
improvement (lower bias and RMSE) over other widely used bulk density equations. The key advantage is that the
present method regresses in relevant physical parameters directly, rather than using discrete bins (for snow class,
elevation, month of year, etc.), each with its own set of model coefficients. The comparison (Figs. 9-10; Table 4) to
the model of Sturm et al. (2010) reveals a peculiar behavior of that model for the Taiga snow class, with a large
negative bias in the Sturm estimates. Inspection of the coefficients provided for that class (Table 3) shows that the
model simply predicts that $\rho_b = \rho_{max} = 0.217$ for all conditions.

When our multi-variable two-equation model, developed solely from western North American data, is applied to
northeast USA locations, it produces SWE estimates with smaller RSME values and larger biases than the western
stations. When comparing the SWE-$h$ curves of the SNOTEL data (Figure 4b) to those of the east coast data sets
(left column; Figure 11), it is clear that the northeast data generally have more scatter. This is confirmed by
computing the correlation coefficients between SWE and $h$ for each dataset. It is unclear if this disparity in
correlation is related to measurement methodology or is instead a 'signal to noise' issue. Comparing Figures 4 and
11 shows the considerable difference in snowpack depth between the western and northeastern data sets. When the
western dataset is filtered to include only measurement pairs where $h < 1.5$ m, the correlation coefficient is reduced
to a value consistent with the northeast datasets. This suggests that the performance of the current (or other)
regression model is not as good at shallow snowpack depths. This is also suggested upon examination of the time
series of observed $\rho_b = SWE/h$ for a given season at a snow pillow site. Very early in the season, when the depths
are small, the density curve is very noisy. Later in the season, when depths are greater, the density curve becomes
much smoother.

When applied to the Chugach coring measurements, the model appears to perform well. The higher values of bias
and RMSE (when compared to Tables 4 and 5) are higher in part since the measurements (and model estimates) of
SWE are only at times of larger snow depth. The variability of the Chugach avalanche probe measurements was
assessed by taking the standard deviation of 8 $h$ measurements per site. The average of this standard deviation over
the sites was 22 cm and the average coefficient of variation (standard deviation normalized by the mean) over the
sites was 15%. Propagating this uncertainty through the regression equations yields a slightly higher (16%)
uncertainty in the SWE estimates. Clearly, this is a function of surface roughness of the underlying terrain.



Backcountry areas characterized by fields and meadows are likely to have smaller coefficients of variation for
ensembles of depth measurements over a small radius. As a result, SWE estimated from a single depth measurement
should be comparatively accurate. Areas of high surface roughness, characterized by crags, rocks and fallen logs
will have large coefficients of variation and larger numbers of depth measurements should be collected and averaged
to provide the best possible SWE estimate.

Measurement precision also affects the construction of a regression model. Upon inspection of the data, it was
observed that the precision of the depth measurements was approximately 25 mm and that of the SWE
measurements was approximately 2.5 mm. To test the sensitivity of the model coefficients to the measurement
precision, the depth values in the training dataset were randomly perturbed by +/- 25 mm and the SWE values were
randomly perturbed by +/- 2.5 mm and the regression coefficients were recomputed. This process was repeated
numerous times and the mean values of the perturbed coefficients were found to be $(A, a_1, a_2, a_3) =$
$(0.0188, 0.9737, 0.2034, 0.4301)$ and $(B, b_1, b_2, b_3) = (0.0386, 0.9535, 0.2745, 0.2184)$. These adjusted
coefficients were then used to recompute the SWE values for the validation data set and the bias and RMSE were
found to be -10.5 mm and 72.7 mm. This represents a roughly 10% increase in RMSE, but a considerable increase in
bias magnitude (see Table 4 for the original values). This sensitivity of the regression analysis to measurement
precision underscores the need to have high-precision measurements for the training data set. It also raises the
interesting question of whether or not future resources should be directed towards expanding networks (greater
spatial coverage) of current technologies or towards refining instrumentation (better accuracy) at currently
instrumented stations.

Another important consideration has to do with the uncertainty of depth measurements that the model is applied to.
For context, one application of this study is to crowd-sourced, opportunistic snow depth measurements from
programs like the Community Snow Observations (CSO; Hill et al., 2018) project. In the CSO program,
backcountry recreational users submit depth measurements, typically taken with an avalanche probe, using a
smartphone in the field. The measurements are then converted to SWE estimates which are assimilated into
snowpack models. These depth measurements are 'any time, any place' in contrast to repeated measurements from
the same location, like snow pillows or snow courses. Most avalanche probes have cm-scale graduated markings, so
measurement precision is not a major issue. A larger problem is the considerable variability in snowpack depth that
can exist over short (meter scale) distances. Recalling the Chugach discussion above, even in flat areas, with a
smooth snow surface (away from major drifting or wind scour), terrain features such as rocks, logs, and vegetation
can produce large variations in probe measurements.

Expansion of CSO measurements in areas lacking SWE measurements can increase our understanding of the
extreme spatial variability in snow distribution and the inherent uncertainties associated with modeling SWE in
these regions. It could also prove useful for estimating watershed-scale SWE in regions like the northeastern USA,
which is currently limited to five automated SCAN sites with historical SWE measurements for only the past two



decades. Additionally, historical snow depth measurements are more widely available in the Global Historical
Climatology Network (GHCN-Daily; Menne et al. 2012), with several records extending back to the late 1800s.
While many of the GHCN stations are confined to lower elevations with shallower snow depths, the broader
network of quality-controlled snow depth data paired with daily GHCN temperature and precipitation measurements
could potentially be used to reconstruct SWE in the eastern US given additional model development and refinement.
**5 Conclusions**
We have developed a new, easy to use method for converting snow depth measurements to snow water equivalent
estimates. The key difference between our approach and previous approaches is that we directly regress in
climatological variables in a continuous fashion, rather than a discrete one. Given the abundance of freely available
climatological norms, a depth measurement tagged with coordinates (latitude and longitude) and a time stamp is
easily and immediately converted into SWE.
We developed this model with data from paired SWE-$h$ measurements from the western United States and British
Columbia. The model was tested against entirely independent data (primarily snow course; some snow pillow) from
the northeastern United States and was found to perform well, albeit with larger biases and root-mean-squared-
errors. The model was tested against other well-known regression equations and was found to perform better.
This model is not a replacement for more sophisticated snow models that evolve the snowpack based on high
frequency (e.g., daily or sub-daily) weather data inputs. The intended purpose of this model is to constrain SWE
estimates in circumstances where snow depth is known, but weather variables are not, a common issue in sparsely
instrumented areas in North America.
**6 Acknowledgements**
Support for this project was provided by NASA (NNX17AG67A). R. Crumley acknowledges support from the
CUAHSI Pathfinder Fellowship. E. Burakowski acknowledges support from NSF (MSB-ECA #1802726).
**7 Data Access**
Numerous online datasets were used for this project and were obtained from the following locations:
1. NRCS Snow Telemetry: https://www.wcc.nrcs.usda.gov/snow/SNOTEL-wedata.html
2. NRCS Soil Climate Analysis Network: https://www.wcc.nrcs.usda.gov/scan/
3. British Columbia Automated Snow Weather Stations:
507          https://www2.gov.bc.ca/gov/content/environment/air-land-water/water/water-science-data/water-data-
508          tools/snow-survey-data/automated-snow-weather-station-data
4. Maine Cooperative Snow Survey: https://mgs-maine.opendata.arcgis.com/datasets/maine-snow-survey-data
5. New York Snow Survey: http://www.nrcc.cornell.edu/regional/snowsurvey/snowsurvey.html
6. Sleepers River Research Watershed. Snow data not available online; request data from contact at:
512          https://nh.water.usgs.gov/project/sleepers/index.htm



7.  Hubbard Brook Experimental Forest: https://hubbardbrook.org/d/hubbard-brook-data-catalog
8.  CONUS PRISM Data: http://www.prism.oregonstate.edu/
9.  British Columbia PRISM Data: http://climatebcdata.climatewna.com/
10. Alaska PRISM Data: https://irma.nps.gov/Portal/

A Matlab function for calculating SWE based on the results is this paper has been made publicly available at Github
(URL provided upon paper acceptance).



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





678 Figure 1: Conceptual sketch of the evolution of snow water equivalent (SWE) over the course of a water year (black

679 line). Also shown is the evolution of SWE with snowpack depth over a water year (red line). Note the hysteresis

680 loop due to the densification of the snowpack.

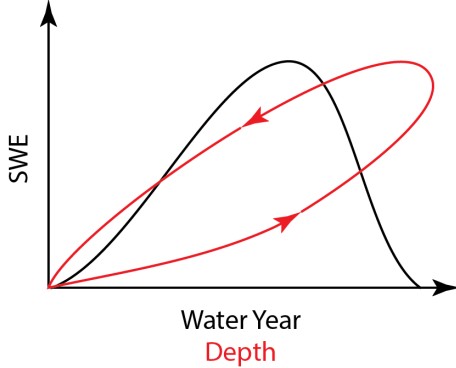




Figure 2: Distribution of measurement locations used in this study.  (a) Western USA and Canada station locations,
with colors indicating station elevation in meters. (b) Northeast USA locations, with stations colored according to
data source. (c, d) Measurement sites in the Chugach Mountains, southcentral Alaska.

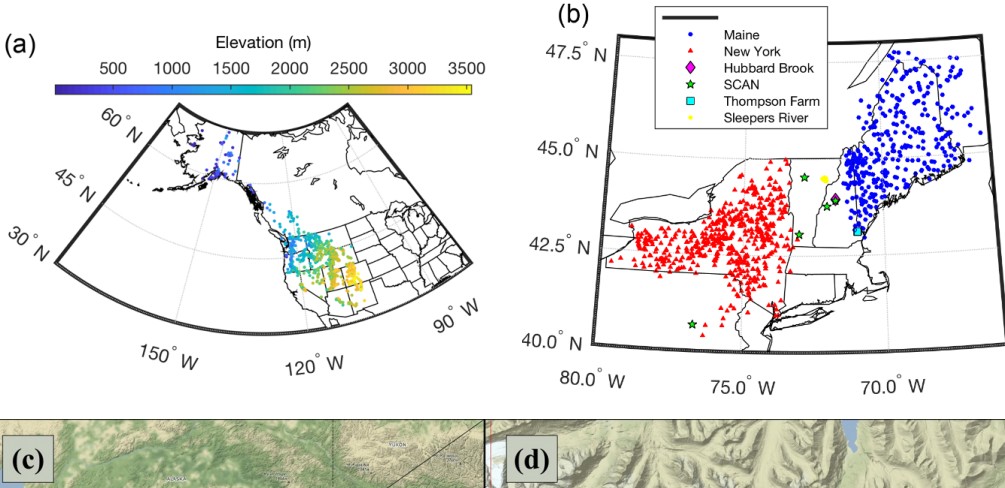


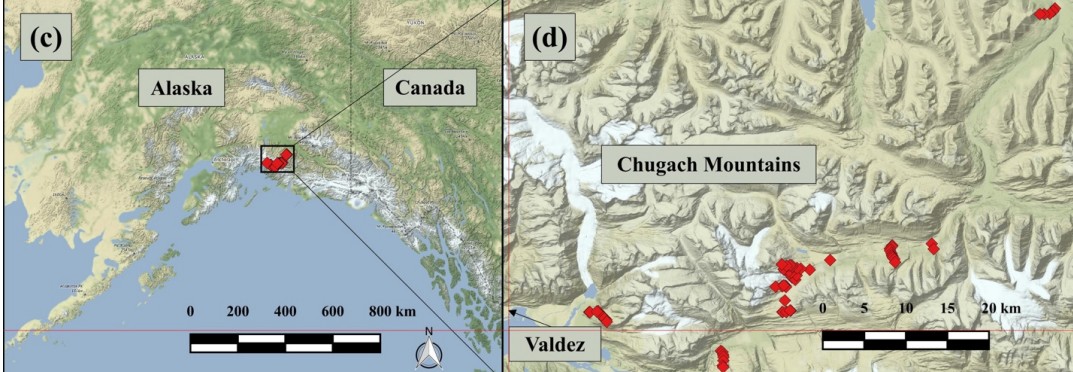







Figure 3: Sample time series of SWE and *h* from the Rex River (WA) SNOTEL station. Observations of *h* at times
when SWE is zero are likely spurious.

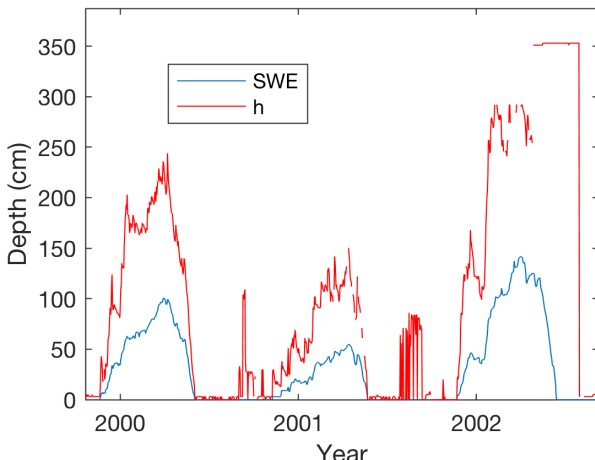






Figure 4: Scatter plot of SWE vs. *h* for the complete SNOTEL dataset before (a) and after (b) removing outliers.
Symbols are colored by 'day of water year' (*DOY*; October 1 is the origin).

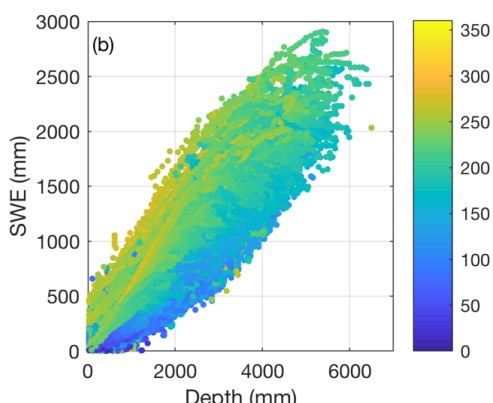




Figure 5: Gridded maps of mean annual precipitation (MAP) and mean February temperature ($\overline{T}_F$) for the study
regions. Climate normals are from the PRISM data set (1981-2010 for CONUS and British Columbia; 1971-2000
for Alaska).




Figure 6: Scatter plots (left column) of modeled vs. observed SWE and probability density functions (right column)
of the residuals for three simple models applied to the CONUS, AK, and BC snow pillow data. Top row (a-b): One-
equation model (Section 2.2.1). Middle row (c-d): Two-equation model (Section 2.2.2). Bottom row (e-f): Multi-
variable two-equation model (Section 2.2.3).

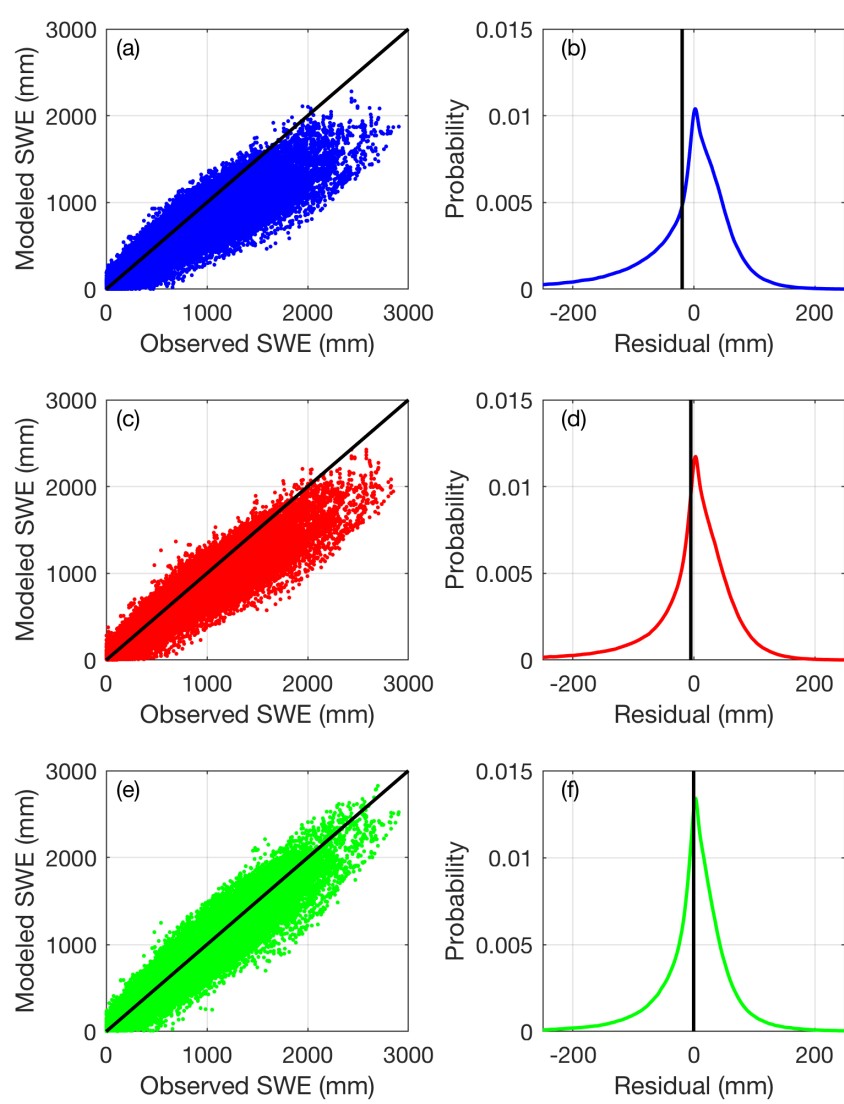






Figure 7: Probability density function of snow pillow station root-mean-square error (RMSE) normalized by station
mean annual precipitation (MAP).

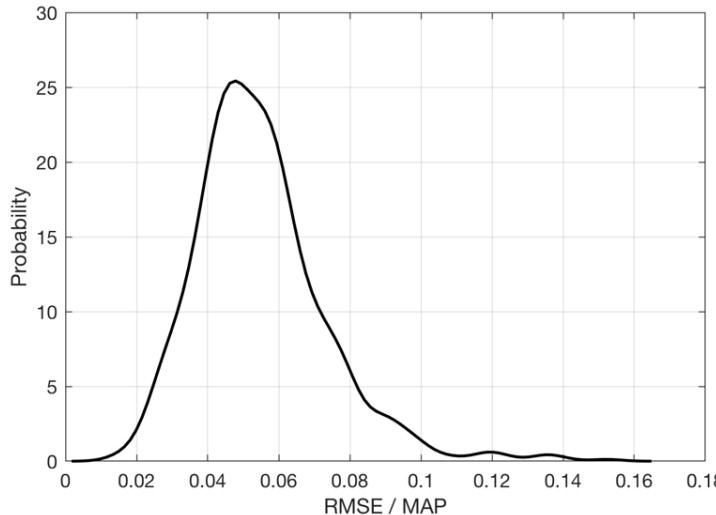






Figure 8: Spatial distribution of RMSE/MAP at snow pillow stations.

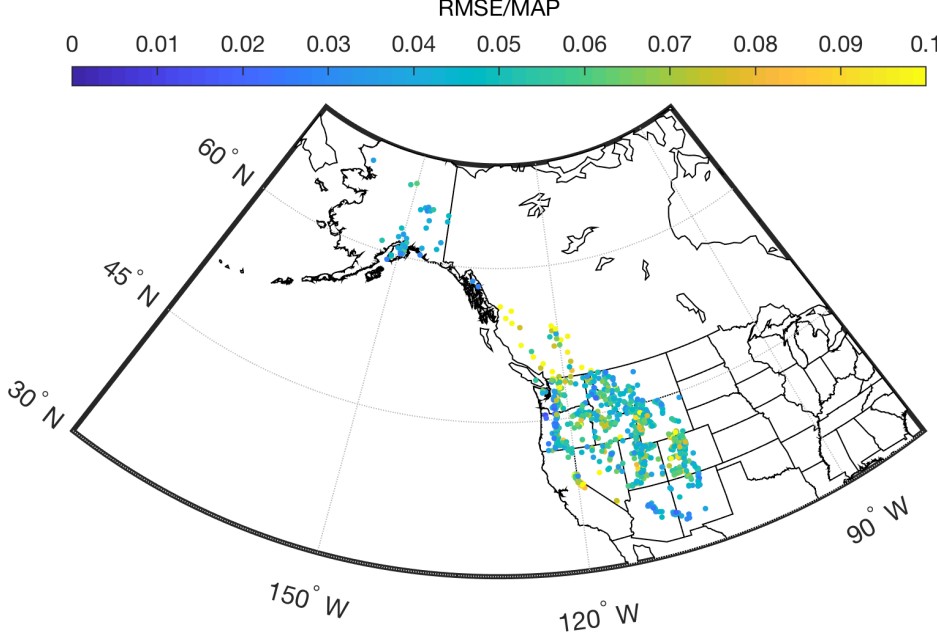






Figure 9: Comparison of the multi-variable, two-equation model of the present study with the model of Sturm et al.
(2010). The subpanels show modeled SWE vs. observed SWE for all of the data binned together, as well as for the
data broken out by the snow classes identified by Sturm et al. (1995). The gray symbols show the Sturm result and
the colored symbols (draped on top) show the current result. The models are being applied to the validation data set
(50% of the aggregated snow pillow data for CONUS, AK, and BC).

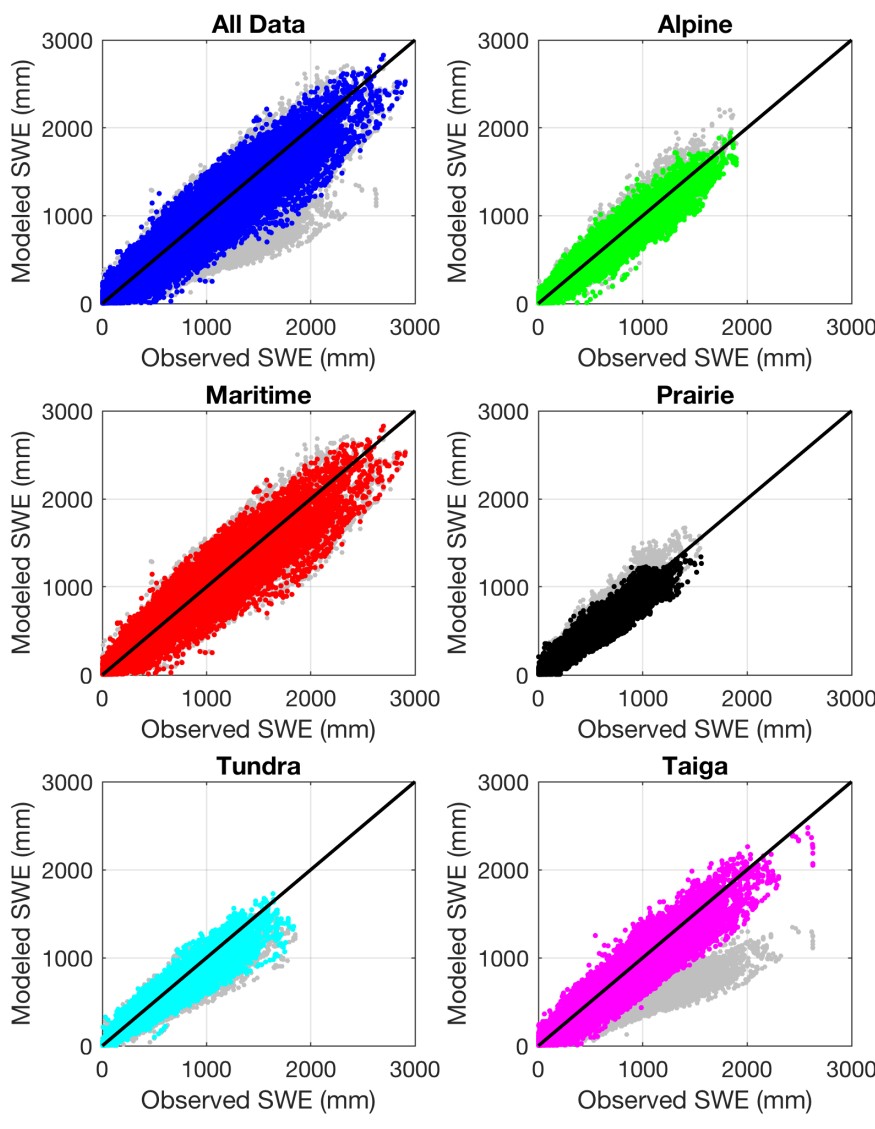




Figure 10: Comparison of the multi-variable, two-equation model of the present study with the model of Sturm et al.
(2010). The subpanels show probability density functions of the residuals of the model fits for all of the data binned
together, as well as for the data broken out by the snow classes identified by Sturm et al. (1995). The gray lines
show the Sturm result and the colored lines show the current result. The vertical lines show the mean error, or the
model bias, for both the Sturm and the current result. The models are being applied to the validation data set (50% of
the aggregated snow pillow data for CONUS, AK, and BC).

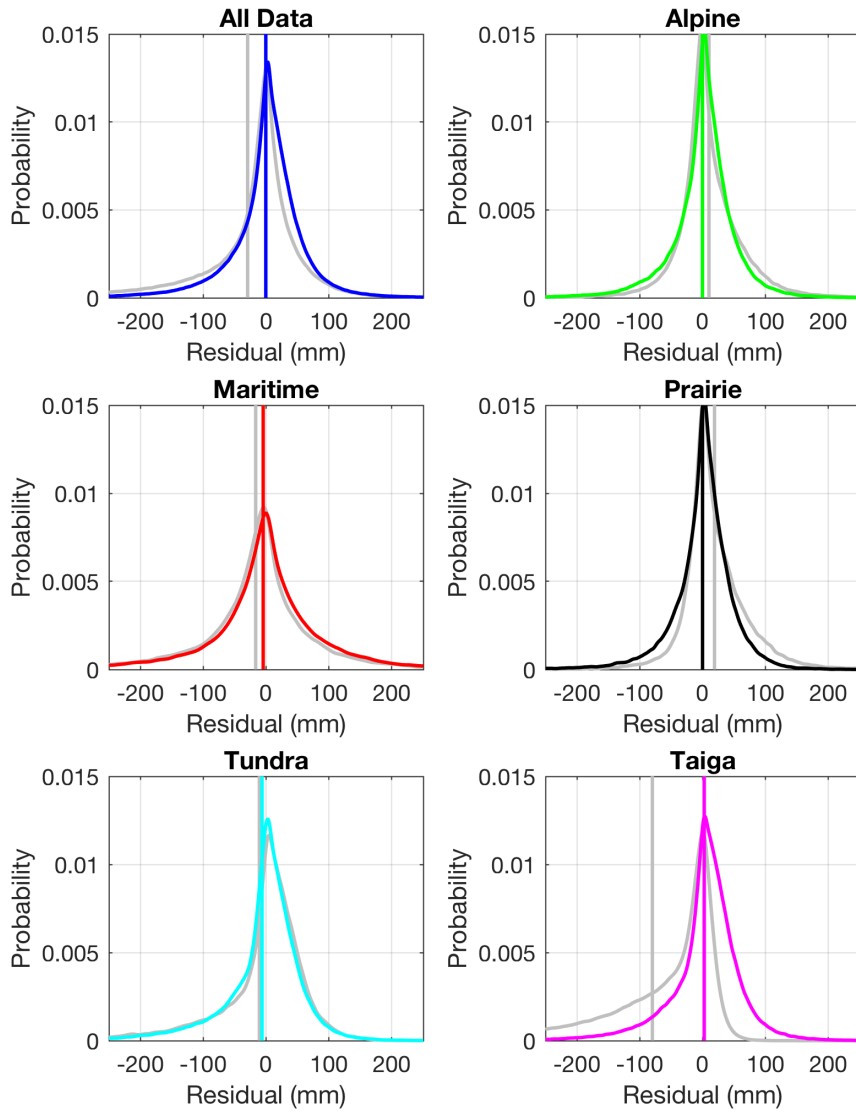






Figure 11: Results from application of the multi-variable, two-equation model to numerous east coast datasets. The
left column shows the SWE-*h* data for each dataset. Note that the black symbols are points removed by the outlier
detection procedure discussed in section 2.1.1.4. The remaining symbols are colored by DOY. The middle panel
plots the model estimates of SWE against the observations of SWE with the 1:1 line included. The right panel shows
probability density functions of the model residuals, with the vertical line indicating the mean error, or bias.
Individual rows correspond to individual data sets and are labeled.

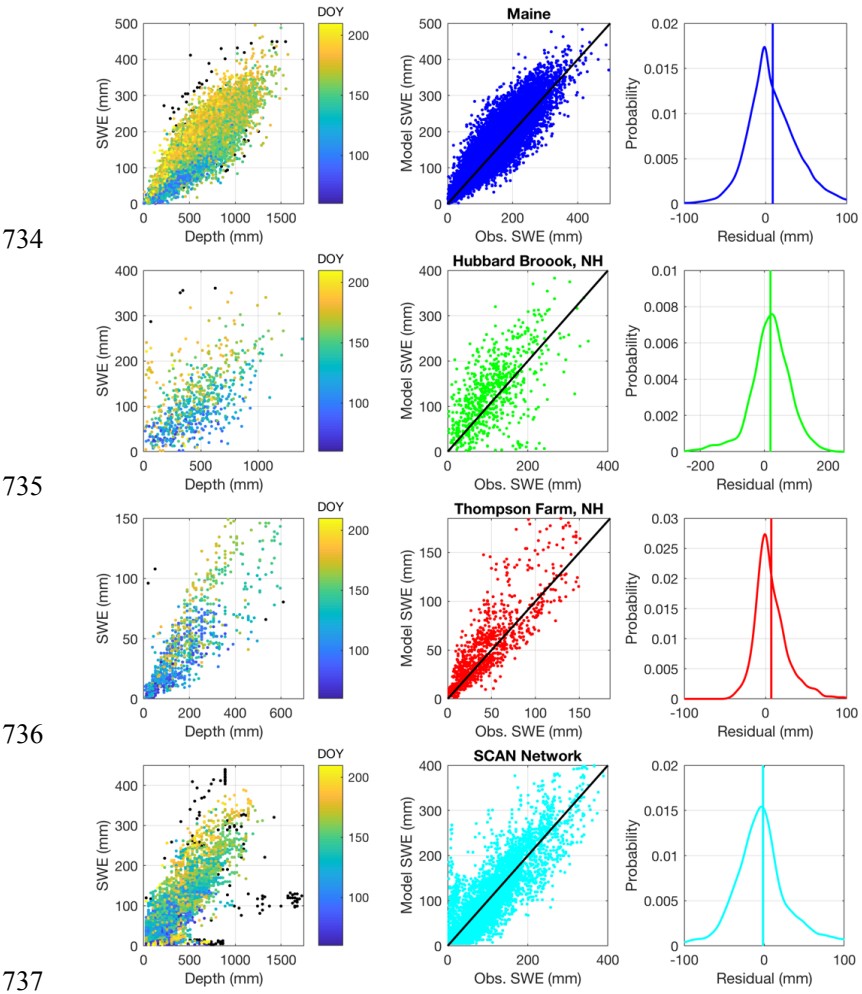







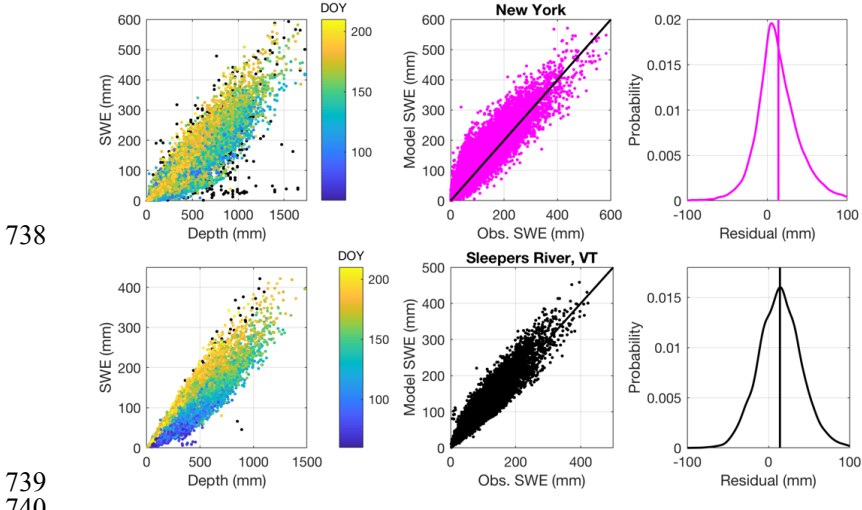







Figure 12: Results from application of the multi-variable, two-equation model to the Chugach Mountains, AK. The
left column shows the measured SWE-*h* data. The symbols are colored by DOY. The middle panel plots the model
estimates of SWE against the observations of SWE with the 1:1 line included. The right panel shows the model
residuals, with the vertical line indicating the mean error, or bias.

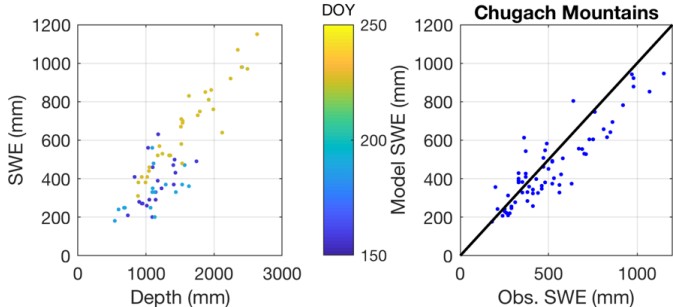
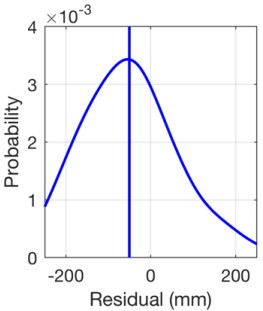
