# Peer review of "Converting Snow Depth to Snow Water Equivalent Using 1 Climatological Variables 2"

_The Cryosphere, 2018_

## Referee Comment (RC1) · Matthew Sturm (Referee) · 21 Feb 2019

Review of Converting Snow Depth to Snow Water 1 Equivalent Using Climatological Variables

February 18, 2019

In this paper, the authors address the problem of converting more readily obtained snow depth measurements to snow water equivalent values. The problem is highly topical as airborne lidar and airborne and satellite-based photogrammetric snow depths become more readily available for widespread use. The authors primarily build on the method described by Sturm, Taras, Liston, Derksen, Jonas and Lea (2010), with the main difference in their method being the replacement of climate classes of snow by

continuous climate variables (mean annual precipitation and February mean temperature) obtained from the PRISM data set. Though not explicitly stated, the authors also establish their regressions using a larger data set than the Sturm et al. study and most other studies of which I am aware. They reach the conclusion that their regressions show an improvement over the 2010 work.

As the lead author of that prior depth-to-SWE study, I find this a fine piece of work, clearly and honestly written, and useful to many practitioners. It should be published. That said, I am not sure that I fully agree with the conclusion of the authors as to the extent of the improvement, whether their improved method is more easily applied than the old, and I find the omission of any discussion of the well-known errors in the data set used to develop the regression equations troubling. I would like to see the authors grapple with this last issue explicitly in the paper before a version of the paper is published.

Examining the input data for this study (Table 1), 98.5% is essential SNOTEL snow pillow data; 1.5% comes from coring. Both types of data are known to contain biases. My personal experience for the latter (coring) is that it tends to undersample SWE (or produce low-biased density values), and across prior studies, there is agreement the method is no more accurate than about $\pm 10\%$. It has been some time since I worked through the literature on snow pillow data, but I recall significant biases from these instruments as well. One source of error is due to snow bridging with, particularly, low biases during the melt when percolating meltwater can run off the pillow to the surrounding snow pack due to the shape of the pillows. Sonic sounders also can exhibit some measurement errors (in this case the ones near the SNOTEL site paired to the pillow SWE values, chiefly in not being representative of the snow depth on the pillow.

Given these potential sources of error, and the fact that the authors are attempting to develop general depth-SWE regressions, they should examine how these errors might cause their results to deviate from the "true" local conversion functions. For example,

hypothetically, in a maritime regime, perhaps the natural snow packs retains frequent rain-on-snow water, but at the actual measurement sites it runs off from the pillows. Then there would be a consistent tendency in this February-warm location with high MAP (mean annual precipitation) to have light (or low) SWE vs. depth values. At least describing in what ways the modeled SWE values might diverge from the on-the-ground values would alert readers to limitations in the methodology.

As far as whether this study is an improvement over our 2010 study, it really comes down to which ancillary data set one wants to work with: a gridded data set of snow classes or of PRISM climate data. Each has advantages and disadvantages in terms of computational cost and hassle. Looking at Table 4 which compares our prior work to the new work, most of the statistical improvement comes from the taiga snow class, which, as the authors note (Line 415), is because in 2010 we assigned a fixed value to this class (e.g., a fixed value performed better than regressed values). This snow class was only 6% of our training set, and I suspect the sample we chose tended to be quite "stiff" because of the high percentage of depth hoar found in taiga snow, thus it did not tend to densify due to overburden stress (probably something of an Arctic bias we showed). The authors taiga sample set is deeper with greater SWE.

One last substantive comment: The authors have an entire section on outlier detection and removal, but I would argue they have potentially removed real data. I applaud them for recognizing the hysteresis loop that is produced by depth-SWE seasonal evolution (Fig. 1) and their clever way of handling it in their regressions (Equation 5). We had actually during our work looked at using a rotated lemniscate to model this behavior, but dropped it because we could not make it work right. But if one recognizes that physically the bulk density increases during the melt during the Spring, then one also has to recognize that very early in the winter, deep fluffy snow will be found on some snow pillows….snow with bulk density values of that are less than 150 kg/m3. Figure 4 (clean version) has a lower depth-SWE line that at 2000 mm is about 350 mmm SWE, a density of 175 kg/m3, and a density of 180 kg/m3 at 3000 mm depth. I belief actual

depth-SWE data on the low end has been removed, not erroneous data. Now one might argue we may in general seem to introduce a low bias when we do these sort of regressions, but that is not reason to label what may be accurate physical data as outliers. As further confirmation, the color of the removed data in Figure 4 is mostly blue (early season) and this removal would impact thin climate classes (e.g. taiga) more than thick classes.

One final comment, and this would be not only to the authors of this paper, but virtually every author out there. Please try to cite the seminal or original papers on a topic if possible. . .not the newest or easiest to cite. The authors here do well in citing Alford and Church, but when it comes to recognizing how snow depth and SWE are related in time and space, the seminal work of G. A. McKay should not be overlooked.

McKay, G. A., and B. F. Findlay, 1971: Variation of snow resources with climate and vegetation in Canada. Proc. 39th Western Snow Conf., Billings, MT, Western Snow Conference, 17–26.

McKay, G. A and D. M. Gray, 1981: The distribution of snowcover. Handbook of Snow, 1st ed. D. M. Gray and E. D. H. Male, Eds., Pergamon Press, 153–190.

Detailed Comments:

Line 36: Surely someone before 2018 recognized that snow was important to hydrology. . .like Gerdel (see U.S. Army Corps of Engineers [1956] monograph on snow hydrology.

Lines 41-50: This paragraph is a little jumbled and doesn't address some of the well-known errors present in snow pillow measurements (see major comments), yet in the next paragraph, errors in SWE core values, which may actually be smaller in some cases, are identified. Perhaps here is where errors in the input data could be discussed in greater detail.

Lines 60-66: This little sections seems uneven, and given the huge literature on trying

to extract SWE from remote sensing, particularly radar, very one-dimensional. Why even talk about snow remote sensing in the paper? I would simply say if falls outside of the scope of the work. . ..and if there really is a reliable operational way to get SWE now from space, I don't know it.

Line 74: Again, Goodison did the seminal work on the sonic sounders. Perhaps you could cite him.

Line 96: I think this citation should be: Jonas, Tobias, Christoph Marty, and Jan Magnusson. "Estimating the snow water equivalent from snow depth measurements in the Swiss Alps." Journal of Hydrology 378, no. 1-2 (2009): 161-167.

Lines 117-118: I do not agree that a priori complexity produces more accuracy. What is really going on here is that proximity to high quality input data tends to produce better accuracy. But that may be true whether the model used with the data is complex or simple. Basically, in a very heterogeneous snow world, when we have local driving data, the results regardless of the model, get better. One might even be able to argue, given the difficulties of measuring radiation in snowy locations, that energy balance models can introduce errors. I don't think you need to work so hard on making a case for the type of statistical approach developed in this paper. Ease of use, and generally the lack of driving data most places, make the case for you.

Line 282: I like this section on DOY, even though in the end you fix the value to 180. Just the fact that the regressions are insensitive to the DOY of peak SWE is interesting.

Lines 336-337: I wish the authors would expand this section. It is the heart of why the regressions work, and it is how this study and our 2010 study are related. Climate classes tell us which snow is warm and deep and tends to densify rapidly; high MAP and high February temperatures tell us the same thing, perhaps as the authors claim, even better (or maybe it is just that the training set being larger is better?) This said, the authors I think are aware that there are several snow packs in which due to development of depth hoar and wind slab, there is very limited increase in density over time.

Icy snow too can resist densificiation.

Line 342: Figure 6 is nice and clear.

Line 357: The model errors will have NO impact on the local snow regime. . .I think you mean the impact will be on the predicted results.

Lines 405 – 410: I realize that the authors are fond of their Chugach results, undoubtedly obtained with much effort, but these data constitute 0.004% of the entire ensemble and could readily be omitted, with the space saved a deeper examination of why the systems is working, and where I might fail to work well.

Lines 431-432: Consider why this is: early in the winter, the addition of new snow to a thin pack makes a dramatic change in the bulk density (e.g., called here noise, but which is real) while later in the winter that noise dissipates because the addition is an increasingly small percentage of what is already on the ground. While a model using historical data cannot adjust for this effect, one could talk about how the uncertainty in the modeled result decreases with time. Does it then increase again after the DOY of peak SWE?

Lines 447-460: This is much too cursory a discussion of precision and accuracy, and it sets up a false strawman: more stations or better precisions? The real question is how do we achieve better accuracy, and by this I suspect we mean better more accurate assessments of snow water resources. Given that 95% of the data being used is SNOTEL measurements, then this question has to start with whether the SNOTEL sites were actually designed to be "representative" or "index" sites. . ..and I believe they were always meant to be the latter. Next it has to proceed to the issue of representativeness, as increasingly as we get depths from lidar or photogrammetry, we will be converting depths to SWE in locations not sampled by the SOTEL network. Are we moving into locations where the bulk density is likely to be higher or lower than at an index station? Why? I would rather see the authors just bypass the issue than trivialize the problem in a statistical experiment that doesn't tell us much about the core issue.

---

## Referee Comment (RC2) · Adam Winstral (Referee) · 5 Mar 2019

The aim of this study is to develop a simple means of estimating snow densities to convert observed snow depths to snow-water-equivalent. The authors seek to use long-term climatological variables rather than station or modeled data so that snow depths garnered in remote locations without direct meteorological observations (e.g. crowd-sourced or Lidar data) can be easily and accurately converted to SWE. There is a growing need for improved means of characterizing snow density as greater amounts of snow depth data are becoming available (e.g. Lidar). Therefore this type of research is certainly warranted. Given that snow depths have always been more readily available than SWE or density data, other researchers have similarly produced methods of estimating densities. While not all of the previously developed approaches tackle the

specific case presented here (i.e. meteorological data immediately preceding snow depth observation not required), the Sturm et al. (2010) and Jonas and Magnusson (2009) approaches do. The authors clearly acknowledge this and make a positive comparison of their method measured against the Sturm method. However, I find the presented Sturm comparison to be biased against the Sturm method (see further comments). They also hint at (lines 413, 493), but never provide evidence nor specifically claim to be, better than the Jonas approach. I would like to see the authors present in a more convincing manner how and why their method represents a substantial advancement over the previously published methods before I am ready to consider this manuscript worthy of publication.

This is my major concern:

The authors randomly split the aggregated CONUS, AK, and BC data into training and validation datasets (Section 2.2). They then use the "held-out" validation dataset to make the Sturm comparison (Section 3.1). So, essentially they have trained their model on data from the same locations with the same statistical metrics present in the comparison dataset. On the other hand, the comparison dataset is 100% independent of the Sturm training data. In order to present a fair comparison this needs to be done with a dataset that is totally independent from the derivation of both. The northeast dataset would be one ideal dataset for conducting this test and I'm not sure why this wasn't done. That said, it would certainly be more convincing if the inter-model comparisons were conducted over a wider range of conditions. I would also like to see direct comparisons to the Jonas method. As I stated in the above paragraph, the authors must present a convincing case that the new methodology represents an improvement over existing procedures. I just don't find that in the current manuscript.

Moderate concerns that need addressing:

I don't understand why rmse was normalized with respect to mean annual precipitation (Section 3 and Figure 8). This obviously biases the normalizations low where summer

precipitation is more common. Artifacts of this can be seen in Figure 8 (e.g. low ratios in Arizona, New Mexico, Alaska where summer precipitation can be considerable compared to winter; high ratios in eastern Sierras where synoptic summer storms are rare). This type of normalization might be appropriate for annual or longer hydrologic studies, but for this snow-based, winter-focused research the normalization should be based on either mean wintertime precipitation or better yet, mean annual snowfall. Both mean wintertime precipitation and mean annual snowfall should be easily derivable from the PRISM data already used in this study.

Graphs. There are way too many data points in the scatter plots to understand what is really going on in Figures 6 and 9, and some of the plots in Figure 11. These should be presented as either heat plots or randomly select and plot a subset of these data. Additionally and partly due the aforementioned reason, the overlapping plots in Figure 9 are impossible to fully discern.

I had difficulty accepting the reasoning for the residuals and mean biases apparent in the Figures 6b and d. I think these residuals, which are present in the validation dataset are also related to the choice of fitting a power law relationship rather than a linear least squares one. Given that the training and validation data should maintain the same statistical metrics then these residuals should be present in the training data as well. If, in fact, this is the case then the combination of a power law fit and the predominance of accumulation season samples would be the reason. My suspicion is that if a linear least squares fit was chosen then there should be near zero mean biases in both the training and validation sets given that the two sets maintain the same characteristics. I would expect that in the linear scenario, there should be a wider spread in residuals (i.e. higher rmse) but very little change in mean bias. Of course, this would be entirely different if the validation set was truly independent.

How the different datasets were used needs better clarification. I didn't understand the purpose of the manually sampled Chugach data. As far as I can tell, these data were not included in the calibration nor the validation analyses. What do these data show?

Why were they included? How do these data add anything new to the analysis? This should be clearly stated and incorporated into the story or leave the Chugach data out.

Section 2.1.2. Do these PRISM climatological variables, based on sparse station data and resolved at 800m, really pick up the heterogeneity you're aiming to capture as expressed on lines 132-37. It would be nice if you could show a spatially explicit example showing these capabilities.

Tidbits:

The residuals (e.g. Figure 6) should be presented as modeled minus observed. In this manner the underestimations of SWE appear as negative residuals rather than the positive residuals currently presented. I find this much easier to understand.

Lines 44-47 and 72-74. Each of these sentences contain two distinct thoughts that would perhaps be better if split into two sentences.

Lines 120-22. I didn't think this sentence was necessary . . . unless you turn it into reasoning that this just adds a layer of computational costs / complexity that aren't necessary for your desired application.

Lines 141-2. Might want to add something about why you would also prefer to not use NWP data that could possibly substitute for the lack of observations (i.e. computational costs, errors in NWP data).

Line 169. You also used snow pillow data from the northeast US. You might want to make that clear here . . . as in "Snow data for this project, aside from the aforementioned SNOTEL data, . . ."

Section 2.1.1.5. Might want to mention that these issues are most common in summer when vegetation grows beneath the sensor.

Line 440. Roughness of underlying terrain is certainly one factor, but couldn't there be others as well (e.g. wind redistribution).

---

## Referee Comment (RC3) · Anonymous Referee #3 · 13 Mar 2019

The authors address the issue of converting spatiotemporal snow depth measurements to estimates of snow water equivalent (SWE). This topic is relevant to many areas of research because of the relative ease of taking snow depth measurements over SWE. Framed in the context of citizen science or field work, snow depths collected by non-experts and experts alike can be leveraged as a low-cost input to hydrological or climate analysis. In an era of high-availability altimetry (lidar or radar) and photogrammetry (structure from motion), an ensemble of methods to convert surface heights into SWE will be critical for both targeted basin studies (ASO) as well as future satellite missions.

The authors develop three regression models to evaluate a snow depth to SWE conversion. Regression skill is evaluated using depth alone, depth separated by accumulation and ablation phases, and depth in combination with climate normal for precipitation,

temperature as well as elevation. Their work differs from previous studies such as Sturm et al. 2010 in that the climate inputs are regressed as continuous variables. As such, any measurement of snow depth with coordinates could potentially be converted, independent of measurement scale. In general the paper to well written and clear in its advancements. The focus on estimates during the ablation phase is a clear contribution, where methods fail. Addressing that 'not all snow is equal' is a strength of the approach.

Prior to publication, I would like to see the outlier detection and validation portions of the paper revisited to reinforce the statistical analysis. While I agree that outlier detection is necessary, an enhanced description of where and when the outliers originate would help to identify potential seasonality or spatial clustering. For example, if many of the outliers are from the early snow season, does this preclude ability of the models to convert measurements that include fresh snow? There are artifacts in Figure 4 where SWE varies drastically but depth does not, are these melt events? A histogram of the outlier DOY or a table of the outlier properties may be all that is needed to address this. These additions could be used to reinforce the statement that the reduced dataset is physically plausible (Lines 229-230). For the validation, it may be of benefit to use a cross-validation (CV) to determine if the model skill is overly optimistic. Using an N-folds CV with a 80/20 train/test split would be a simple approach to achieve this. In this regard, I'd also be interested to know if the non-SNOTEL datasets actually influence to the regression coefficients (What happens when the training datasets are SNOTEL only). The remainder of my comments addressed to specific lines or figure.

Lines 64-65: Are there additional references available to support this statement regarding L-band? The only cited application in the field is a conference proceeding.

Line 172: Each style of corer has its own associated bias. Could this be considered to bound or constrain errors for each region/dataset?

Line 185: I would expect readers to be unfamiliar with some of coring devices. For

example, the Mt. Rose snow tube could be supported with Church, J. Improvement in Snow Survey Apparatus, TAGU, 1936.

Line 228: See concerns about outlier detection in the main comments. It would be important to describe the temporal aspect of the outlier detection.

Line 228: uncleaned data -> source data

Line 229: State how many outliers were removed from the other datasets via this process. Figure 4: An axis label is needed for the DOY color bar.

Line 231: How does this work for 'stations' where there are a very low number of observations, ie AK?

Table 1: Can this table be augmented with a % of retained points or an omission %? Is the BC survey missing the # of ultrasonic sites?

Line 250: Is this 50% of all measurements or 50% of each subset. If it is all of them, it could be such that the only ones removed are CONUS because of the low numbers elsewhere.

Line 256: Figure 3 is used as support for the outlier detection due to poor correlation (ie increasing h with no SWE) and but is referenced here as strongly correlated. It might be confusing to do both.

Line 283: If this is an important consideration, why is the SCAN dataset not used to train the models?

Line 290: Interesting that a static 180 works best as the DOY separator. Could a sentence on why this might occur be added to the discussion?

Line 332: I see how it would not be possible to use an absolute value here but are snow-covered regions where the February normal is below -30C.

Figure 6: Titles for each plot might make this easier to read if someone skips the

caption.

Table 5: Include the normalized errors for completeness of the table.

Line 423-430: Might be helpful to discuss measurement errors as a contributor.

---

## Author Comment (AC1) · 11 Apr 2019

**Reply to Matthew Sturm, Referee**

Review of Converting Snow Depth to Snow Water 1 Equivalent Using Climatological Variables

February 18, 2019

Referee comments are left-justified, in black. Author replies are indented, in blue.

In this paper, the authors address the problem of converting more readily obtained snow depth measurements to snow water equivalent values. The problem is highly topical as airborne lidar and airborne and satellite-based photogrammetric snow depths become more readily available for widespread use. The authors primarily build on the method described by Sturm, Taras, Liston, Derksen, Jonas and Lea (2010), with the main difference in their method being the replacement of *climate classes of snow* by continuous climate variables (*mean annual precipitation and February mean temperature*) obtained from the PRISM data set. Though not explicitly stated, the authors also establish their regressions using a larger data set than the Sturm et al. study and most other studies of which I am aware. They reach the conclusion that their regressions show an improvement over the 2010 work.

As the lead author of that prior depth-to-SWE study, I find this a fine piece of work, clearly and honestly written, and useful to many practitioners. It should be published. That said, I am not sure that I fully agree with the conclusion of the authors as to the extent of the improvement, whether their improved method is more easily applied than the old, and I find the omission of any discussion of the well-known errors in the data set used to develop the regression equations troubling. I would like to see the authors grapple with this last issue explicitly in the paper before a version of the paper is published.

Thank you for this general overall assessment. Below, we provide point-by-point responses to your comments and we indicate where and how we plan to revise the manuscript prior to publication.

Examining the input data for this study (Table 1), 98.5% is essential SNOTEL snow pillow data; 1.5% comes from coring. Both types of data are known to contain biases. My personal experience for the latter (coring) is that it tends to undersample SWE (or produce low-biased density values), and across prior studies, there is agreement the method is no more accurate than about ±10%.

We would like to point out that all of the data used to construct the regression model are snow pillow data. Table 1 summarizes the data used to build the model, and also the other independent data sets used to validate the model. We have clarified in the manuscript (beginning of section 2.2) and table (using bold font in the table) which data are used for what purposes.

It has been some time since I worked through the literature on snow pillow data, but I recall significant biases from these instruments as well. One source of error is due to snow bridging with, particularly, low biases during the melt when percolating meltwater can run off the pillow to the surrounding snowpack due to the shape of the pillows. Sonic sounders also can exhibit some measurement errors (in this case the ones near the SNOTEL site paired to the pillow SWE values, chiefly in not being representative of the snow depth on the pillow.

> This is an important point. There are some studies[1] that show that SNOTEL sites can report SWE > accumulated precipitation, attributed to drifting snow. However, this would not bias snow density assuming that the SWE and Hs measurements are co-located. There are other studies[2] that have looked at the measurement bias in SWE depending on whether or not the snow pillow is steel vs. hypalon. One comprehensive study[3] of biases notes a complex situation, where SWE is sometimes under-reported due to 'snow bridging', but over-reported at other times (see Table 1 of that paper). While that paper proposes methods for correcting SWE measurements, it is complex in practice, requiring continuous SWE, Hs, and near-ground temperature measurements. Please continue to our next remark below.

Given these potential sources of error, and the fact that the authors are attempting to develop general depth-SWE regressions, they should examine how these errors might cause their results to deviate from the "true" local conversion functions. For example, hypothetically, in a maritime regime, perhaps the natural snow packs retains frequent rain-on-snow water, but at the actual measurement sites it runs off from the pillows. Then there would be a consistent tendency in this February-warm location with high MAP (mean annual precipitation) to have light (or low) SWE vs. depth values. At least describing in what ways the modeled SWE values might diverge from the on-the-ground values would alert readers to limitations in the methodology.

> This is a sensible suggestion. In the first draft of the paper, we did investigate the effect of measurement precision. In our revision, we now provide more discussion about potential errors in snow pillow measurements (to help alert readers, as you suggest). One complicating issue is that many studies that report on 'errors' in SWE from snow pillows define this error as the difference between the snow pillow and a coring measurement. The implicit assumption is that the coring measurement is the 'ground truth' but as you note, coring is good to +/- 10%. Given the lack of any consensus information on the distributions of errors in snow pillow measurements (we provide some citations to show the divergence of studies out there), we are unable to provide any good quantitative information on the effects of the pillow errors on the SWE estimates.

[1] https://journals.ametsoc.org/doi/pdf/10.1175/JHM-D-12-066.1

[2] https://www.nrcs.usda.gov/Internet/FSE_DOCUMENTS/nrcs141p2_032059.pdf

[3] https://onlinelibrary.wiley.com/doi/abs/10.1002/hyp.5795

As far as whether this study is an improvement over our 2010 study, it really comes down to which ancillary data set one wants to work with: a gridded data set of snow classes or of PRISM climate data. Each has advantages and disadvantages in terms of computational cost and hassle. Looking at Table 4 which compares our prior work to the new work, most of the statistical improvement comes from the taiga snow class, which, as the authors note (Line 415), is because in 2010 we assigned a fixed value to this class (e.g., a fixed value performed better than regressed values). This snow class was only 6% of our training set, and I suspect the sample we chose tended to be quite "stiff" because of the high percentage of depth hoar found in taiga snow, thus it did not tend to densify due to overburden stress (probably something of an Arctic bias we showed). The authors taiga sample set is deeper with greater SWE.

> With regards to relative model performance. In our first draft, we tried to be as objective and factual as possible, in the sense of simply providing the comparative results (both figures and RMSE values). We feel that this is fair and appropriate. We also felt it appropriate to break out results by snow class so that readers could see how the comparison varied based on that. We have added information about how many data points in the aggregated CONUS, BC, and AK dataset are in each snow class (in section 3.1).

> Regarding computational cost and hassle. The Sturm approach uses a straightforward equation and only requires access to the 1km snow-cover raster. Our approach uses more 'data' in the sense that numerous PRISM grids are required. However, we have packaged all necessary files into a freely available (will be released on GitHub upon acceptance of this paper) function that is very easy to use. By doing so, we alleviate any cost and hassle concerns.

One last substantive comment: The authors have an entire section on outlier detection and removal, but I would argue they have potentially removed real data. I applaud them for recognizing the hysteresis loop that is produced by depth-SWE seasonal evolution (Fig. 1) and their clever way of handling it in their regressions (Equation 5). We had actually during our work looked at using a rotated lemniscate to model this behavior, but dropped it because we could not make it work right. But if one recognizes that physically the bulk density increases during the melt during the Spring, then one also has to recognize that very early in the winter, deep fluffy snow will be found on some snow pillows....snow with bulk density values of that are less than 150 kg/m$^3$. Figure 4 (clean version) has a lower depth-SWE line that at 2000 mm is about 350 mmm SWE, a density of 175 kg/m3, and a density of 180 kg/m3 at 3000 mm depth. I belief actual depth-SWE data on the low end has been removed, not erroneous data. Now one might argue we may in general seem to introduce a low bias when we do these sort of regressions, but that is not reason to label what may be accurate physical data as outliers. As further confirmation, the color of the removed data in Figure 4 is mostly blue (early season) and this removal would impact thin climate classes (e.g. taiga) more than thick classes.

> This is a great point, and one which the authors have discussed at some length, following your review. Manual examination of many of the SNOTEL time series revealed

the presence of clearly wrong data (Figure 3 of the paper). We wanted to develop a wholly objective method for removing those data points. There is a lack of clarity and/or consensus in the literature about how to do this. The approach that we used seems like a good one, based upon the characteristics of the bivariate distribution. We recognize that some valid data points (mostly at low SWE-Hs values) are undoubtedly removed as well. Given the very low number (less than 1%; so the valid points removed are some small fraction of this 1%) of points that were removed in our process, we feel that this is acceptable.

This figure of the output of the data removal process illustrates things. SWE on vertical axis, h on horizontal. Removed points are in red

[Figure]

We acknowledge that referring to this process as 'outlier' detection is perhaps too strong and we have modified the language accordingly, notably re-titling Subsection 2.1.1.5. We also note that Anonymous Referee #3 had a similar comment and wanted to see a histogram of the DOY of the removed data points. We have gone back and looked at the distribution of DOY for all removed points. It turns out that the mean value of DOY was 160 and the standard deviation was 65. So, the bulk of the removed points comes from the middle of the snow season, not at the beginning or the end. This seems to alleviate a bit of the concern that you raise above.

One final comment, and this would be not only to the authors of this paper, but virtually every author out there. Please try to cite the seminal or original papers on a topic if possible…not the newest or easiest to cite.  The authors here do well in citing Alford and Church, but when it comes to recognizing how snow depth and SWE are related in time and space, the seminal work of G. A. McKay should not be overlooked.

McKay, G. A., and B. F. Findlay, 1971: Variation of snow resources with climate and vegetation in Canada. Proc. 39th
Western Snow Conf., Billings, MT, Western Snow Conference, 17–26.

McKay, G. A and D. M. Gray, 1981: The distribution of snowcover. Handbook of Snow, 1st ed. D. M. Gray and E. D. H. Male, Eds., Pergamon Press, 153–190.

We have added the former citation.

Detailed Comments:

Line 36: Surely someone before 2018 recognized that snow was important to hydrology…like Gerdel  (see U.S. Army Corps of Engineers [1956] monograph on snow hydrology.

This citation has been added.

Lines 41-50: This paragraph is a little jumbled and doesn't address some of the well-known errors present in snow pillow measurements (see major comments), yet in the next paragraph, errors in SWE core values, which may actually be smaller in some cases, are identified. Perhaps here is where errors in the input data could be discussed in greater detail.

Yes, we have reworked the introduction a fair bit to bring in some information upfront about errors in coring and in snow pillows.

Lines 60-66: This little sections seems uneven, and given the huge literature on trying to extract SWE from remote sensing, particularly radar, very one-dimensional. Why even talk about snow remote sensing in the paper?  I would simply say if falls outside of the scope of the work….and if there really is a reliable operational way to get SWE now from space, I don't know it.

We were trying to be comprehensive in laying out all of the options (in-situ vs. remote) for acquiring snow information. Your suggestion (huge literature that we don't do justice to) is on point and we have removed this section from the paper.

Line 74: Again, Goodison did the seminal work on the sonic sounders. Perhaps you could cite him.

This is a sensible suggestion and we will do so.

Line 96:  I think this citation should be: Jonas, Tobias, Christoph Marty, and Jan Magnusson. "Estimating the snow water equivalent from snow depth measurements in the Swiss Alps." Journal of Hydrology 378, no. 1-2 (2009): 161-167.

Correct. We had it right in the references, but incorrect in the in-line citation.

Lines 117-118: I do not agree that *a priori* complexity produces more accuracy. What is really going on here is that proximity to high quality input data tends to produce better accuracy. But that may be true whether the model used with the data is complex or simple. Basically, in a very heterogeneous snow world, when we have local driving data, the results regardless of the model, get better. One might even be able to argue, given the difficulties of measuring radiation in snowy locations, that energy balance models can introduce errors. I don't think you need to work so hard on making a case for the type of statistical approach developed in this paper. Ease of use, and generally the lack of driving data most places, make the case for you.

> This is a fair point and we revised our wording.

Line 282: I like this section on DOY, even though in the end you fix the value to 180. Just the fact that the regressions are insensitive to the DOY of peak SWE is interesting.

> Agreed, thank you for noting this. It was an unexpected result.

Lines 336-337: I wish the authors would expand this section. It is the heart of why the regressions work, and it is how this study and our 2010 study are related. Climate classes tell us which snow is warm and deep and tends to densify rapidly; high MAP and high February temperatures tell us the same thing, perhaps as the authors claim, even better (or maybe it is just that the training set being larger is better?) This said, the authors I think are aware that there are several snow packs in which due to development of depth hoar and wind slab, there is very limited increase in density over time. Icy snow too can resist densificiation.

> We have expanded this section somewhat. We fully recognize that all bulk-density methods that rely on simple inputs like DOY or climatological weather characteristics are unable to capture numerous features of snowpacks. That is a limitation of the emphasis on simplicity.

Line 342: Figure 6 is nice and clear.

> Thank you. We have slightly modified this to show the data clouds as heat maps (2d histograms, essentially) at the suggestion of another reviewer.

Line 357: The model errors will have NO impact on the local snow regime...I think you mean the impact will be on the predicted results.

> Correct, this was poorly stated, and we have reworded this.

Lines 405 – 410: I realize that the authors are fond of their Chugach results, undoubtedly obtained with much effort, but these data constitute 0.004% of the entire ensemble and could readily be omitted, with the space saved a deeper examination of why the systems is working, and where I might fail to work well.

> Another referee was also lukewarm on the inclusion of this dataset. We have removed most of it, except for the useful information that it provides on the variability in Hs that is observed over short distances. That is a valuable point to retain.

Lines 431-432: Consider why this is: early in the winter, the addition of new snow to a thin pack makes a dramatic change in the bulk density (e.g., called here noise, but which is real) while later in the winter that noise dissipates because the addition is an increasingly small percentage of what is already on the ground. While a model using historical data cannot adjust for this effect, one could talk about how the uncertainty in the modeled result decreases with time. Does it then increase again after the DOY of peak SWE?

> These are good points. Yes, our model is using only climatological weather data, which know nothing about individual snowfall events. We have added remarks on this issue and we have added a new figure that shows the errors as a function of the DOY.

Lines 447-460: This is much too cursory a discussion of precision and accuracy, and it sets up a false strawman: more stations or better precisions? The real question is how do we achieve better accuracy, and by this I suspect we mean better more accurate assessments of snow water resources. Given that 95% of the data being used is SNOTEL measurements, then this question has to start with whether the SNOTEL sites were actually designed to be "representative" or "index" sites….and I believe they were always meant to be the latter. Next it has to proceed to the issue of representativeness, as increasingly as we get depths from lidar or photogrammetry, we will be converting depths to SWE in locations not sampled by the SOTEL network. Are we moving into locations where the bulk density is likely to be higher or lower than at an index station? Why? I would rather see the authors just bypass the issue than trivialize the problem in a statistical experiment that doesn't tell us much about the core issue.

> We have removed the commentary on whether or not future investments should be in more stations vs. better stations. We agree that SNOTEL stations are largely index stations in that their measurements are often directly regressed against downstream streamflow. However, for our purposes here (providing an equation to estimate SWE from h 'anywhere anytime') we do feel that it is valuable to discuss the effects of source data accuracy and precision on the estimated SWE. This will help the reader to understand how much uncertainty there will be in SWE derived from the current paper. So, we retain some of the statistical testing.

---

## Author Comment (AC2) · 11 Apr 2019

**Reply to Adam Winstral, Referee**

Referee comments are left-justified, in black. Author replies are indented, in blue.

The aim of this study is to develop a simple means of estimating snow densities to convert observed snow depths to snow-water-equivalent. The authors seek to use long-term climatological variables rather than station or modeled data so that snow depths garnered in remote locations without direct meteorological observations (e.g. crowd-sourced or Lidar data) can be easily and accurately converted to SWE. There is a growing need for improved means of characterizing snow density as greater amounts of snow depth data are becoming available (e.g. Lidar). Therefore this type of research is certainly warranted. Given that snow depths have always been more readily available than SWE or density data, other researchers have similarly produced methods of estimating densities. While not all of the previously developed approaches tackle the specific case presented here (i.e. meteorological data immediately preceding snow depth observation not required), the Sturm et al. (2010) and Jonas and Magnusson (2009) approaches do. The authors clearly acknowledge this and make a positive comparison of their method measured against the Sturm method. However, I find the presented Sturm comparison to be biased against the Sturm method (see further comments). They also hint at (lines 413, 493), but never provide evidence nor specifically claim to be, better than the Jonas approach. I would like to see the authors present in a more convincing manner how and why their method represents a substantial advancement over the previously published methods before I am ready to consider this manuscript worthy of publication.

This is my major concern:
The authors randomly split the aggregated CONUS, AK, and BC data into training and validation datasets (Section 2.2). They then use the "held-out" validation dataset to make the Sturm comparison (Section 3.1). So, essentially they have trained their model on data from the same locations with the same statistical metrics present in the comparison dataset. On the other hand, the comparison dataset is 100% independent of the Sturm training data. In order to present a fair comparison this needs to be done with a dataset that is totally independent from the derivation of both.

> This is an important point. Our current approach aggregated all western North America snow pillow data (some ~2M points) and then randomly split it in two. So, for each station, some data at each station was used for model building, the other data at each station ended up being used for validation. We can see why it would be important to test a validation approach that separated the training and validation data either by location, or by time.

> To address your concerns, we took all of the snow pillow data and we split up the stations randomly into two groups. We took all of the data from the first group and we used that to train the regression model. We then validated the regression model against the second group. We did several realizations of this process and found that the results were extremely close to those presented in the original manuscript. Anonymous

Referee #3 also raised a similar concern, and suggested an 80/20 cross validation (80% of the data used to train, 20% of the data used to validate) approach. This method also generated similar results. We believe this to be due to the very large N of our dataset.

Given how similar all of these approaches were, and given the lack of any clear 'preferred method' in the literature, we decided to retain our original approach.

We strongly agree with the referee that it would be ideal to have a perfect test between the two methods (our model and that of Sturm et al.). However, that would require that the two models be developed with the same training datasets and then validated using the same validation datasets. Unfortunately, we don't see a way to create this perfect 'laboratory test' for two models developed with different data.

The northeast dataset would be one ideal dataset for conducting this test and I'm not sure why this wasn't done. That said, it would certainly be more convincing if the inter-model comparisons were conducted over a wider range of conditions.

You are correct in that it would be ideal to have inter-model comparisons over a wide range of conditions. We believe that applying both models to the NE data set would not accomplish that. We prefer to keep our inter-model comparisons to the larger dataset from western North America snow pillow data, and we will retain the NE dataset for our model only.

I would also like to see direct comparisons to the Jonas method. As I stated in the above paragraph, the authors must present a convincing case that the new methodology represents an improvement over existing procedures. I just don't find that in the current manuscript.

With regards to Jonas et al. (J09). We specifically chose not to apply that model for the following reason. The J09 model has coefficients that depend upon month of year and elevation. In addition to this, there is a geographic 'offset' term that depends on boundaries drawn in the Swiss Alps. Therefore, the model cannot be applied in other regions (since we would have no idea what to use for an offset). We do not wish to ignore the offset and apply a 'partial model' since that is not what those authors constructed.

One thing that we have done is to apply the very simple Pistocchi[1] model which depends only on day of year (DOY). In Pistocchi's paper, he claims comparable performance to both Sturm and J09. We now include summary results (RMSE and bias only, no figures) for the Pistocchi model applied to the western North America snow pillow data.

We believe that the results for our model demonstrate an improvement (lower bias and RMSE than existing methods) and also a strength of our approach is that it allows for a
* * *
[1] https://www.sciencedirect.com/science/article/pii/S2214581816300131

continuously varying snow density in space rather than discontinuities due to discrete snow classes. Our plots below, provided in response to another comment, help illustrate this point.

Moderate concerns that need addressing:
I don't understand why rmse was normalized with respect to mean annual precipitation (Section 3 and Figure 8). This obviously biases the normalizations low where summer precipitation is more common. Artifacts of this can be seen in Figure 8 (e.g. low ratios in Arizona, New Mexico, Alaska where summer precipitation can be considerable compared to winter; high ratios in eastern Sierras where synoptic summer storms are rare). This type of normalization might be appropriate for annual or longer hydrologic studies, but for this snow-based, winter-focused research the normalization should be based on either mean wintertime precipitation or better yet, mean annual snowfall. Both mean wintertime precipitation and mean annual snowfall should be easily derivable from the PRISM data already used in this study.

> This is a reasonable suggestion. Our intent was simply to provide some sort of 'relative' measure of the magnitude of the RMSE. We have actually redone this using the mean annual peak SWE to normalize the RMSE, which makes good sense.

Graphs. There are way too many data points in the scatter plots to understand what is really going on in Figures 6 and 9, and some of the plots in Figure 11. These should be presented as either heat plots or randomly select and plot a subset of these data. Additionally and partly due the aforementioned reason, the overlapping plots in Figure 9 are impossible to fully discern.

> With regards to Figure 11 (Fig 12 in the revision), the symbols are colored by DOY in the left column, so we are unable to show that column as a heatmap. We have changed the center column to show the data as a heatmap.

> With regards to Figure 6, we have changed the plot to a heatmap (which is just a 2d histogram). The 'footprint' or 'envelope' of the data cloud is unchanged of course.

> With regards to Figure 9 (Fig 10 in the revision). The important point is how the 'width' of the data cloud is different between the two methods. The envelope that is closer to the 1:1 line indicates better performance. Our original approach was chosen since, in each case, our envelope was narrower (so we plotted ours on top). We cannot show two overlapping heatmaps. What we have done in the revision is to show Sturm's results as scatter symbols (as before) and to then plot our results as a transparent heat map on top.

I had difficulty accepting the reasoning for the residuals and mean biases apparent in the Figures 6b and d. I think these residuals, which are present in the validation dataset are also related to the choice of fitting a power law relationship rather than a linear least

squares one. Given that the training and validation data should maintain the same statistical metrics then these residuals should be present in the training data as well. If, in fact, this is the case then the combination of a power law fit and the predominance of accumulation season samples would be the reason. My suspicion is that if a linear least squares fit was chosen then there should be near zero mean biases in both the training and validation sets given that the two sets maintain the same characteristics. I would expect that in the linear scenario, there should be a wider spread in residuals (i.e. higher rmse) but very little change in mean bias. Of course, this would be entirely different if the validation set was truly independent.

> This is a fair comment, and our initial remarks may have been too speculative. We adopted a power law relationship based on the hysteresis loop (Figs 1 and 4) suggesting something other than a linear relationship between h and SWE. We feel that the best course of action is to remove our overly speculative comment.

How the different datasets were used needs better clarification. I didn't understand the purpose of the manually sampled Chugach data. As far as I can tell, these data were not included in the calibration nor the validation analyses. What do these data show? Why were they included? How do these data add anything new to the analysis? This should be clearly stated and incorporated into the story or leave the Chugach data out.

> Two reviewers of this paper noted this. We have essentially dropped that dataset from the paper, with one exception. The large ensemble (80 or so) of collections (8 at each site) of probe measurements is valuable since it helps to quantify the variability in snow depth over small distances (in discussion section).

Section 2.1.2. Do these PRISM climatological variables, based on sparse station data and resolved at 800m, really pick up the heterogeneity you're aiming to capture as expressed on lines 132-37. It would be nice if you could show a spatially explicit example showing these capabilities.

> We feel that the continuously variable PRISM data does a better job of capturing climate than 5 snow classes. Let us illustrate this with some sample figures. First, consider the map of snow class in the region just northeast of Valdez, Alaska.

[Figure]

Note that there are only a few snow classes and that the landscape is dominated by class 7 in this case. Now, for the exact same lat / lon bounding box, let us look at the MAP and Feb_T_Mean:

[Figure]

[Figure]

In both of these climatological rasters, we see very considerable variation over a region that is monolithic in snow class. These, we do feel that the use of 800m PRISM data will allow for smoother variability in snow density.

Tidbits:
The residuals (e.g. Figure 6) should be presented as modeled minus observed. In this manner the underestimations of SWE appear as negative residuals rather than the positive residuals currently presented. I find this much easier to understand.

Actually, the residuals are done correctly. Please see the new version of Fig 6, which has been much improved by showing it as a heatmap. Look at the top row. The residuals are indeed computed as model-observed. The vertical black line in the right column (panel (b)) is the mean residual. It is negative. And that makes sense since the cloud of data points appears to be, on average, below the 1:1 line. So, thank you for your suggestion, it was good for us to double check, but we do have the residuals defined correctly, we believe.

Lines 44-47 and 72-74. Each of these sentences contain two distinct thoughts that would perhaps be better if split into two sentences.

Thank you for the suggestion. We will improve the clarity of these lines.

Lines 120-22. I didn't think this sentence was necessary . . . unless you turn it into reasoning that this just adds a layer of computational costs / complexity that aren't necessary for your desired application.

We slightly adjusted the sentences there to improve the clarity.

Lines 141-2. Might want to add something about why you would also prefer to not use NWP data that could possibly substitute for the lack of observations (i.e. computational costs, errors in NWP data).

The purpose of this work is to provide a rapid, easy to use tool. Relying on external daily or sub-daily datasets and/or model output moves the work away from that goal and towards more sophisticated snow models. So, yes, it could be done, but at significant expense and effort.

Line 169. You also used snow pillow data from the northeast US. You might want to make that clear here . . . as in "Snow data for this project, aside from the aforementioned SNOTEL data, . . ."

Yes, thank you. We fixed this.

Section 2.1.1.5. Might want to mention that these issues are most common in summer when vegetation grows beneath the sensor.

We are not sure we fully understand this remark. Which particular issues are you referring to? The data that we considered was only winter time data, where snow was present.

Line 440. Roughness of underlying terrain is certainly one factor, but couldn't there be others as well (e.g. wind redistribution).

We have now noted this explicitly.

---

## Author Comment (AC3) · 11 Apr 2019

**Reply to Anonymous Referee #3**

Referee comments are left-justified, in black. Author replies are indented, in blue.

The authors address the issue of converting spatiotemporal snow depth measurements to estimates of snow water equivalent (SWE). This topic is relevant to many areas of research because of the relative ease of taking snow depth measurements over SWE. Framed in the context of citizen science or field work, snow depths collected by nonexperts and experts alike can be leveraged as a low-cost input to hydrological or climate analysis. In an era of high-availability altimetry (lidar or radar) and photogrammetry (structure from motion), an ensemble of methods to convert surface heights into SWE will be critical for both targeted basin studies (ASO) as well as future satellite missions. The authors develop three regression models to evaluate a snow depth to SWE conversion. Regression skill is evaluated using depth alone, depth separated by accumulation and ablation phases, and depth in combination with climate normal for precipitation, temperature as well as elevation. Their work differs from previous studies such as Sturm et al. 2010 in that the climate inputs are regressed as continuous variables. As such, any measurement of snow depth with coordinates could potentially be converted, independent of measurement scale. In general the paper to well written and clear in its advancements. The focus on estimates during the ablation phase is a clear contribution, where methods fail. Addressing that 'not all snow is equal' is a strength of the approach.

Prior to publication, I would like to see the outlier detection and validation portions of the paper revisited to reinforce the statistical analysis. While I agree that outlier detection is necessary, an enhanced description of where and when the outliers originate would help to identify potential seasonality or spatial clustering. For example, if many of the outliers are from the early snow season, does this preclude ability of the models to convert measurements that include fresh snow? There are artifacts in Figure 4 where SWE varies drastically but depth does not, are these melt events? A histogram of the outlier DOY or a table of the outlier properties may be all that is needed to address this. These additions could be used to reinforce the statement that the reduced dataset is physically plausible (Lines 229-230).

> These are very good points. One other referee had similar remarks. Manual examination of many of the SNOTEL time series revealed the presence of clearly wrong data (Figure 3 of the paper). We wanted to develop a wholly objective method for removing those data points. The approach that we used seems like a good one, based upon the characteristics of the bivariate distribution. We recognize that some valid data points (mostly at low SWE-Hs values) are undoubtedly removed as well. Given the very low number (less than 1%; so the valid points removed are some small fraction of this 1%) of points that were removed in our process, we feel that this is acceptable. Here is a figure of the process at one particular station. SWE on vertical axis, h on horizontal. Red circles are removed points.

[Figure]

We particularly like your suggestion of looking at the characteristics of the removed points, and now include specific information on the DOY values of these points. It turned out that removed points were occurring throughout the snow season, and not just at the beginning and the end.

Your comment about events in Figure 4 where SWE changes a lot while Hs remains fixed is an interesting one. It is hard to understand how SWE could drop from 1 m to near zero while Hs remains fixed at 5 m. The lack of an accepted and easy to implement protocol for addressing snow pillow data quality control is an obstacle to analysis.

For the validation, it may be of benefit to use a cross-validation (CV) to determine if the model skill is overly optimistic. Using an N-folds CV with a 80/20 train/test split would be a simple approach to achieve this. In this regard, I'd also be interested to know if the non-SNOTEL datasets actually influence to the regression coefficients (What happens when the training datasets are SNOTEL only). The remainder of my comments addressed to specific lines or figure.

A few comments. First of all, the regression coefficients were constructed with snow pillow data only from the western United States and Canada. We have now tried to make this more obvious in the section discussing datasets. For example, in Table 1, we now use bold font to highlight which datasets were used to build the model.

Second, with regards to validation. We looked into this at some length before beginning this work, since we wanted to determine if there was some preferred way of doing validation in the snow density literature (or streamflow prediction, or any other discipline for that matter). We found no 'best' or 'preferred' method. We ended up doing a 50/50 split (aggregating all snow pillow data points and randomly dividing them up) in the first draft of the paper. Upon receiving the manuscript reviews, we also tested your suggested 80/20 split, and a 50/50 'station split' (divide up the stations, not the

aggregated data points). We found that all methods provided essentially the same results We feel that this is likely due to the large N (number of observations) of our dataset. Given the lack of consensus in the literature, we feel that our approach is acceptable, and we are clear and upfront about our methods.

Lines 64-65: Are there additional references available to support this statement regarding L-band? The only cited application in the field is a conference proceeding.

Another referee though we should simply remove the remote sensing discussion and that seemed like a sensible change to us, so we did so.

Line 172: Each style of corer has its own associated bias. Could this be considered to bound or constrain errors for each region/dataset?

Corer data were not used to build the regression model. So, those biases would not affect the regression model coefficients. Any depth measurement that has a bias or random error and that is used to estimate a SWE value using the methods in this paper would propagate through into a bias or error in the SWE. We do try to present some discussion on this in the manuscript.

Line 185: I would expect readers to be unfamiliar with some of coring devices. For example, the Mt. Rose snow tube could be supported with Church, J. Improvement in Snow Survey Apparatus, TAGU, 1936.

Thank you for this suggestion, and we can add this citation.

Line 228: See concerns about outlier detection in the main comments. It would be important to describe the temporal aspect of the outlier detection.

Yes, as noted above, we provide information on this now in that section.

Line 228: uncleaned data -> source data

Good catch, we made this change.

Line 229: State how many outliers were removed from the other datasets via this process. Figure 4: An axis label is needed for the DOY color bar.

This has been handled with added parenthetical notes to column 4 of Table 1.

Line 231: How does this work for 'stations' where there are a very low number of observations, ie AK?

The process was objectively applied to all stations. Stations with low numbers of observations could still be processed, in terms of computing the characteristics of the bivariate distributions and then removing points that did not satisfy the criteria.

Table 1: Can this table be augmented with a % of retained points or an omission %? Is the BC survey missing the # of ultrasonic sites?

As per the remark just above re:line 229, yes we have done this. Regarding the BC comment. The first row of that table has two sets of numbers. One for the Western USA SNOTEL. One for the eastern USA SCAN. The BC row only has one set of numbers since we grouped all BC snow pillows together. In revised Table 1, we have split up the USA NRCS data into two rows to eliminate this confusion.

Line 250: Is this 50% of all measurements or 50% of each subset. If it is all of them, it could be such that the only ones removed are CONUS because of the low numbers elsewhere.

All of the aggregated snow pillow data were grouped (data points were grouped in one large bin) and then divided in two. Given the random nature of the division, each station should have ~50% of its data represented.

Line 256: Figure 3 is used as support for the outlier detection due to poor correlation (ie increasing h with no SWE) and but is referenced here as strongly correlated. It might be confusing to do both.

This is a good catch. We meant to refer to just the winter (snow present) portions of Figure 3. The noisy bits in that Figure are at times when there is no SWE. We will clarify our language.

Line 283: If this is an important consideration, why is the SCAN dataset not used to train the models?

There are several reasons. Foremost, we wanted to leave the northeastern USA data alone so that we could use those data as an independent test of the ability of the model to work in completely different regions / snow regimes. Second, the N (5 sites) of the northeastern USA dataset is a tiny fraction of the rest of the available data. Locations with multi-peak SWE curves may do better with a more complex model that is able to capture this behavior.

Line 290: Interesting that a static 180 works best as the DOY separator. Could a sentence on why this might occur be added to the discussion?

To be frank, we do not have a great explanation for this. When we discovered a fairly strong correlation between day of peak SWE and April temperature, we were confident

that the variable DOY approach would produce the best results. In this case, it appears that simpler is better.

Line 332: I see how it would not be possible to use an absolute value here but are snow-covered regions where the February normal is below -30C.

> We chose this offset value based upon the lowest February temperature values observed at the snow pillow stations. This may limit our methods to not apply in some extremely cold regions.

Figure 6: Titles for each plot might make this easier to read if someone skips the caption.

> We appreciate this stylistic suggestion. Our approach favors using the figure caption to provide details on the content in each figure panel, which is consistent with the approach of other papers in The Cryosphere. We are open to modifying this if the editors request it.

Table 5: Include the normalized errors for completeness of the table.

> We are not able to normalize the errors for these datasets in the way that we do for the snow pillow sites (Figs 8-9 of new version of paper). For the snow pillow stations, we normalized the RMSE at each station based on (this is a change for the 2nd draft of this paper) the mean annual maximum SWE at that station. The information in table 5 is different. The RMSE values there are essentially being averaged 'spatially' over a distributed dataset, rather than being averaged temporally at a snow pillow station. Thus, we do not have a mean annual maximum SWE available for normalization in a consistent fashion. Note that in the 2nd paragraph of the discussion section, we do talk a bit about the east coast results and how they differ from western North America (smaller snowpack, etc.).

Line 423-430: Might be helpful to discuss measurement errors as a contributor.

> We do discuss this (measurement errors) in lines 447 → 472 (numbering of original draft). In the specific context of the northeastern USA data, those data are generally high-quality coring data. Having not taken those data ourselves, it is hard to quantify the measurement errors. In some cases, the supporting documentation for those datasets is brief to non-existent. Also, note that, in response to another reviewer as well, we have added more general discussion of both coring error and snow pillow errors to the manuscript.

---

## Author Response (AR2)

**Response to Editor (TC-2019-1)**

Thank you for the handling of our manuscript. Below are private comments to you. We also include a response to A. Winstral. In addition, we include a 'track changes' and also a 'clean' version of our revision. We believe that we have handled all necessary revisions and that we have clearly demonstrated the improvement of our method. We hope to hear from you soon.

Comments to the Author:
Dear authors
Thanks for submitting the revised version of your manuscript. You have addressed all points in your replies raised by the reviewers. However, I believe you have failed to incorporate the arguments you bring forward to actually support your conclusions. I find it not sufficient to mention in the reply that you could have done this or that, or you would have even tried it, but it would not have changed the results.

In our first revision, we made numerous additions. Also, we tested several ways of building the regression model, and we found the results to be the same. So, perhaps there was a miscommunication. It was not that we could have tried a new approach. We actually did a new approach, but it did not make a difference. We state this explicitly in the revised paper.

I expect you to take the arguments by reviewer #2 more seriously and make the appropriate changes to the manuscript. I think it is not too difficult and you are actually close to a valuable contribution that includes a comprehensive, objective analysis with statistically supportable claims. If you are prepared to revise the manuscript accordingly I am happy to reconsider it for publication.

In the latest version you will find a side-by-side comparison of our model to three other widely used models. We apply all four models to three datasets:
1. The western North America snow pillow dataset (close to 2 million data points).
2. The complete western North America snow course / aerial marker data set (100,000+ data points).
3. The northeast USA collection of data sets (100,000+ measurements).
Our model has lower bias and RMSE in all cases, except for two very small northeast USA datasets (totaling about 1500 data points). We believe that this is a very convincing case that our approach is an improvement over existing methods.

Jürg Schweizer

**Response to A. Winstral Review of TC-2019-1**

> Thank you for the recent review of our manuscript. We appreciate the investment of time that you have made in improving our analysis. Below, we provide point-by-point responses to the May 2019 review of our second draft. Reviewer comments are in black. Our responses are indented and in blue. All references to table/figure/line numbers refer to the 'clean' version of revision #2. One notable change is that we have revised our regression analysis to use 'winter precipitation' and 'winter temperature' rather than mean annual precipitation and mean February temperature. This reduced our RMSE values.

Though I firmly stated that the authors had not presented convincing evidence to support their claims, they chose to not implement and include any of my major suggestions for revisions to make the work more convincing. Instead they have largely chosen to reason away my suggestions. I don't wish to repeat my review nor get into a lengthy debate on the relevancy of my comments. Instead I'll just go with what the presented data say to me. The authors have derived a method for estimating snow densities that includes climatological variables, which provide a means of capturing spatial heterogeneity. They developed and tested this method using data from snow pillows and snow courses. By the authors own admissions (lines 46-50) these data all come from "relatively simple topography". Generally these are all located in flat, wind-sheltered locations. On the other hand the Sturm model, a well-regarded and oft-cited research piece, is based on similar data as well as data collected on manual traverses representing a range of topographic positions and snow deposition zones. The presented comparison with the Sturm model is conducted only at SNOTEL sites (i.e. flat, wind-sheltered). These same exact sites were used in both the calibration of the new model and for comparisons with the Sturm model. Results show that overall the new model performs better. However, if one were to eliminate taiga sites, which were not well represented in the Sturm data, the overall results are very similar (see table below).

> We do not believe this to be an entirely correct statement. In Table 4 of the paper, we provide comparative results for 'all data' and also for the data broken out by snow class. In every row of this table, the results show that our model has a smaller bias and RMSE than Sturm's model.

> Note also that, in the revision, we now compare our model to Sturm at all of the northeastern USA data (nearly all snow course, not snow pillow). Our model has lower bias and RMSE than Sturm for all of these data sources. We also now compare our model to Sturm at all (100,000+ measurements) NRCS snow course sites in western North America. These sites are independent from the model training data and use a different method (snow coring instead of snow pillow). Our model has lower bias and RMSE than Sturm for this independent data set.

Though the authors acknowledged that I raised an important point regarding their splitting of data in which all stations were included in both training and validation sets they state only that a station-dependent splitting method produced "extremely close" results to the original without actually presenting those results.

Yes, this was a good point, and we tested it out after your first review. The results were the same. It felt redundant to us to present numeric data (identical) for both of these cases in the revised manuscript. In our new revision, we now explicitly state (beginning of section 2.2) that we did our model building / validation both ways, and that the results were the same.

Given how similar the provided results are for non-taiga performance of the two models, the only conclusion I can draw from the presented data is that at sheltered, taiga sites the new model performs better. At other sheltered locations, the new model might be better or worse.

Table 4 shows that the current model has smaller biases and RMSEs for all snow classes, not just Taiga. For the non-taiga sites, our RMSE is usually ~20% smaller than Sturm. And, our biases are much smaller than the biases of Sturm.

Additionally, the new application of our model, side-by-side with Sturm for two independent data sets shows that our model consistently has lower RMSE and bias.

The new method is trained in the same conditions and sites as the validation was conducted whereas the Sturm method is based on a greater diversity of data from independent sites. Given this unbalanced methodology and the closeness of the results, I find the comparative assessments of model performance at sheltered, non-taiga sites to be uncertain.

See our above remarks. We believe the modifications we have made alleviate this point, which is similar to several above points.

Results averaged over all snow classes except taiga (taken from snow class percentages provided in Section 3.1 and results in Table 4)
Sturm et al. Multi-variable two-equation
R2 0.97 0.97
rmse (mm) 72.1 67.8
bias (mm) 1.76 -2.26

Regarding the usefulness and accuracy of the new methodology at sites other than typical SNOTEL stations for estimating densities nothing has been shown (e.g. the manuscript-referenced crowd-sourced and Lidar data that are gathered in a variety of topographic settings). No data is presented for anything other than flat, wind-sheltered locations therefore conclusions on model performance in any other conditions are not possible.

We showed in our original paper our model applied a great variety of data (snow course, mostly) from the northeastern USA. The northeastern US snow courses (NY and ME Snow Survey sites) are not typical SNOTEL stations and include both forested and open sites in both flat and topographically complex settings. At Hubbard Brook, the snow courses are forested areas about ¼ hectare in size. Thompson Farm in Durham, NH, USA includes a forested site and open site. Sleeper's River in Vermont also includes forested and open field sites across a range of topographic classes.

In addition to this, we have now included all western North America snow course data (100,000+ measurements). These are completely different data from the SNOTEL snow pillow data.

So, we have great confidence in the ability of our model to perform in a wide variety of environments.

Yes, the maps look pretty and capture greater heterogeneity but how accurate are they? A theoretical argument can easily be made that the Sturm method – based on a more diverse set of data – would actually be better.

I would be remiss if I didn't further comment on one of my major suggested revisions that wasn't followed up on: direct comparisons to the Jonas et al. method. Rather than including direct comparisons to the Jonas method (based on over 11,000 observations and cited over 140 times) as suggested, the authors have chosen to make comparisons to the simple Pistocchi method that is solely a function of day-of-year (based on 206 observations and cited 5 times). According to the authors this was necessary because the Jonas method is dependent on month of year, elevation, and a geographic "offset" term. Certainly the former two variables are available to the authors. The presence of the offset term however leads the authors to conclude that the Jonas model cannot be applied to other regions while implying that Jonas et al. did not "construct" their model for such applications. Yet in referring to the importance of the offset term Jonas et al. state, "However, the minor importance of regional effects suggests that the model may also be applicable in other regions with similar snow climatologic conditions." In fact, if one averaged the regional offset term over all the data records in the Jonas application it comes to a mere 3 kg/m3. One could in fact, easily optimize the Jonas "offset" term to the data presented here in the calibration set. (It is my personal belief that any comprehensive analysis should include this). Yet even a straight-out-of-the-box Jonas application using the average offset or none at all would be insightful. If, in fact, the presented model performs better than the Jonas model at the tested sites then this would show that there are indeed regional constraints present in the Jonas model that must be accounted for. On its own, these insights on regionality for one of the most widely cited density parameterizations would be relevant. Of course, this would also lend greater credence to the model presented in this work as well. This entire analysis including optimization could probably be done in less than a day. Without optimizing the offset term, this is about 5 lines of code that could even be handled in a spreadsheet. I truly don't understand why this revision wasn't undertaken.

We now present results for the Jonas model applied to (1) the SNOTEL data, (2) the northeastern USA data, and (3) all NRCS snow course data. Please see sections 3.3, 3.4, and 3.5. The present model has lower bias and RMSE than Jonas for (1), for four out of 6 datasets in (2) and for (3). The two datasets for (2) where Jonas has lower RMSE and bias are the smallest datasets by 1-2 orders of magnitude.

Unfortunately, save for the inclusion of the Pistocchi model comparison, the authors have presented no new data. It is my opinion that the presented work still does not compellingly support the stated conclusions, particularly this one (lines 432-33), "The results presented in this study show that the regression equation described by equations (5, 7-8) is an improvement (lower bias and RMSE) over other widely used bulk density equations."

> The present model is now compared against three others (Sturm, Jonas, and Pistocchi) for many different datasets from different regions and different methodologies. The results objectively show that the present model has lower values for bias and RMSE.

The authors have the data available to make this a much more extensive and scientifically supportable work (e.g. significance tests would be a nice touch).

> We are not 100% sure which specific tests you are asking for. We have provided metrics such as RMSE, R2, and bias. If you are referring to significance tests related to the linear regression analysis, the p values for each variable were 0, and we have now stated that in the manuscript. We also (this was discussed in the original paper and the discussion remains) indicate the use of adjusted R2 values in the construction of the model. This is important since one can always add more predictors and improve the R2. The adjusted R2 ensures that we are only adding in more predictors that produce enough of an improvement in R2 to justify the inclusion.

I think I have provided several constructive ideas on how to go about this. At the very least, if they weren't to follow up on these suggestions they need to objectively evaluate their results and in my opinion, substantially scale back their claims.

> In summary, our model is compared to three others for three different datasets, representing a wide variety of methodologies (for measuring SWE) and physical locations.
> 1. Western North America snow pillow data
> 2. Western North America snow course data
> 3. Northeast USA data (mostly snow course, some snow pillow)
>
> For all three of these datasets, our model had the smallest errors and biases.

[revised manuscript text omitted]